# PKCδ is an activator of neuronal mitochondrial metabolism that mediates the spacing effect on memory consolidation

**Typhaine Comyn, Thomas Preat, Alice Pavlowsky\*, Pierre-Yves Plaçais\***

Energy & Memory, Brain Plasticity Unit, CNRS, ESPCI Paris, PSL Research University, Paris, France

## eLife Assessment

This is a **fundamental** research study which identifies some of the molecular mechanisms underlying the energy costly process of memory consolidation. The strength of evidence is **exceptional**. The paper should be of broad interest because it establishes a clear mechanistic link between long-term memory processes and the energy-producing machinery in neurons.

**\*For correspondence:**
alice.pavlowsky@espci.fr (AP);
pierre-yves.placais@espci.fr (P-YP)

**Competing interest:** The authors declare that no competing interests exist.

**Abstract** Relevance-based selectivity and high energy cost are two distinct features of long-term memory (LTM) formation that warrant its default inhibition. Spaced repetition of learning is a highly conserved cognitive mechanism that can lift this inhibition. Here, we questioned how the spacing effect integrates experience selection and energy efficiency at the cellular and molecular levels. We showed in *Drosophila* that spaced training triggers LTM formation by extending over several hours an increased mitochondrial metabolic activity in neurons of the associative memory center, the mushroom bodies (MBs). We found that this effect is mediated by PKCδ, a member of the so-called 'novel PKC' family of enzymes, which uncovers the critical function of PKCδ in neurons as a regulator of mitochondrial metabolism for LTM. Additionally, PKCδ activation and translocation to mitochondria result from LTM-specific dopamine signaling on MB neurons. By bridging experience-dependent neuronal circuit activity with metabolic modulation of memory-encoding neurons, PKCδ signaling binds the cognitive and metabolic constraints underlying LTM formation into a unified gating mechanism.

## Introduction

Long-term memory (LTM) is a fundamental cognitive process that allows organisms to efficiently retain and retrieve information over extended (and sometimes lifelong) periods of time. Since the long-term retention of fortuitous or poorly relevant associations can drive maladaptive behavior, the triage of learning experiences and routing to LTM has to be tightly gatekept. As a result of this cognitive constraint, brains have evolved mechanisms for the default inhibition of LTM formation (*Abel et al., 1998*; *Scheunemann et al., 2018*). Under certain circumstances, LTM inhibition is lifted in a process known as LTM gating. Defective gating in either direction is expected to cause memorization deficits or (on the contrary) hypermnesia, with both conditions leading to a severe deterioration in the quality of life. It is therefore crucial to decipher the underlying mechanisms behind this precisely regulated process. One of the most conserved conditions known to potently lift the default inhibition of LTM formation is the spaced repetition of learning (*Abel et al., 1998*;

*Shaughnessy, 1977*), which has been reported in a number of species from *Aplysia* (*Sutton et al., 2002*), *C. elegans* (*Nishijima and Maruyama, 2017*) and *Drosophila* (*Tully et al., 1994*) to mice (*Glas et al., 2021*) and humans (*Shaughnessy, 1977*). This phenomenon is called the spacing effect, in contrast with intensive learning (or cramming), in which massed presentation of information leads to the formation of a less persistent form of consolidated memory (*Shaughnessy, 1977*). Studies in animal models, in particular in fruit flies using an associative aversive olfactory paradigm, demonstrated that LTM formation following spaced training exerts a significant energetic burden on organisms (*Mery and Kawecki, 2005*; *Padamsey and Rochefort, 2023*; *Plaçais et al., 2017*; *Plaçais and Preat, 2013*). Following the paired delivery of an odorant with electric shocks, flies show learned avoidance of this odorant for a couple hours (*Quinn et al., 1974*). Whereas spaced repetition of odor/shock pairing allows sustained retention (up to 1 week; *Bouzaiane et al., 2015*; *Heisenberg, 2003*), massed training results in the rapid decline (1–2 days) of the formed memory, which provides a powerful, experimentally tractable way to model the spacing effect versus cramming. Strikingly, in the hours following spaced training in fruit flies, their sucrose intake is observed to increase by over twofold (*Plaçais et al., 2017*), which does not occur after massed training. In addition, spaced, but not massed, training decreases the survival duration of flies under conditions of limited resources (*Plaçais and Preat, 2013*). These observations reveal a major impact of LTM formation on the energy balance of the whole organism, with this metabolic constraint being an additional incentive for the default inhibition of LTM formation. Interestingly, the recent finding that cellular energy fluxes are involved in the control of LTM formation (*Plaçais et al., 2017*) suggests that both cognitive and metabolic constraints linked with LTM formation may be merged in a unitary, cost-effective gating process. Indeed, we previously showed that the upregulation of mitochondrial metabolic activity in neurons of the mushroom bodies (MB), the insect brain's memory center, in the first hours following spaced training is critical to initiating LTM formation (*Plaçais et al., 2017*). This upregulation of mitochondrial metabolism depends on post-learning dopamine signaling from a specific pair of MB-afferent neurons, called MP1 neurons (or PPL1-γ1pedc neurons *Aso et al., 2014*), which show sustained calcium rhythmic activity in the same time windows encompassing the first hours after spaced training (*Plaçais et al., 2012*). MP1 activates MB mitochondrial activity via the DAMB receptor (also named Dop1R2; *Plaçais et al., 2017*). The putative interdependence of cognitive and metabolic constraints, two sides of the same coin, therefore calls for the identification of the neuronal molecular gatekeeper at the core of the spacing effect, linking neuronal network activity and mitochondrial energy metabolism.

The G-protein-coupled receptor DAMB preferentially engages with Gq (*Cassar et al., 2015*; *Himmelreich et al., 2017*), which signals via the second messengers DAG (*Mizuno and Itoh, 2009*) and calcium, two canonical activators of a broad family of serine/threonine kinases called PKC enzymes. Within the PKC family, two subfamilies can be activated by downstream effectors of Gq: the classical PKCs (PKCα, PKCβ1 and β2, PKCγ), that are regulated by both DAG and calcium, and the novel PKCs (PKCδ, PKCε, PKC η , PKCθ,), that are exclusively activated by DAG but not calcium (*Duquesnes et al., 2011*). In *Drosophila*, genes of the two PKC subfamilies are present (*Shieh et al., 2002*): the classical PKCs PKC53E and eye-PKC, the novel PKCs PKC98E and PKCδ, as well as a PKC-related kinase (CG2049). Their expression patterns in the fly brain have not been systematically characterized, however single-cell transcriptomic data show that all of them are expressed at various levels across the *Drosophila* brain (*Davie et al., 2018*). Among those PKC isoforms, PKCδ shows unique properties that make it a candidate of particular interest for mediating DAMB signaling. In non-neuronal mammalian cells, it was shown that PKCδ can translocate to mitochondria upon its activation (*Wu-Zhang et al., 2012*), where it specifically localizes to the intermembrane space. There, PKCδ is able to activate oxidative metabolism by targeting the pyruvate dehydrogenase (PDH) complex (*Acin-Perez et al., 2010*; *Kim and Hammerling, 2020*), which catalyzes the first step of pyruvate metabolism for oxidative phosphorylation. Despite this evidence, the mechanisms underlying the regulation of PKCδ translocation to mitochondria upon activation, and how it interfaces with extracellular signals, remains poorly understood. Although PKCδ expression is commonly used to identify specific neurons in the mammalian brain (*Cai et al., 2014*; *Cui et al., 2017*; *Dilly et al., 2022*; *Haubensak et al., 2010*; *Wang et al., 2020*; *Williford et al., 2023*), the functional role of PKCδ in brain tissues has remained thus far largely unexplored. Nonetheless, the putative ability of PKCδ to act as the interface between Gq-mediated activation and metabolic activity control prompted us to study this kinase as the missing piece

in the puzzle of LTM gating that could bridge dopaminergic activation and neuronal mitochondrial metabolism.

We first uncovered that PKCδ activation is required in MB neurons for LTM formation through behavior experiments involving targeted genetic inhibition of PKCδ and in vivo functional brain imaging using a specific sensor of PKCδ activity, δCKAR (*Kajimoto et al., 2010*; *Wu-Zhang et al., 2012*), which we introduced in *Drosophila*. We further showed that PKCδ is a downstream effector of DAMB and that its translocation to the mitochondria is triggered upon LTM formation by MP1/DAMB signaling. Employing in vivo imaging of cellular pyruvate consumption, we showed that PKCδ intervenes in LTM formation through its metabolic role as a mitochondrial activator, by releasing the pyruvate dehydrogenase (PDH) complex inhibition. We therefore revealed that a major effect of spaced training is to mobilize the DAMB/PKCδ signaling cascade in order to perpetuate for several hours a neuronal metabolic enhancement that only transiently occurs after a single learning session. Overall, our data establish PKCδ as an essential activator of neuronal pyruvate mitochondrial metabolism that mediates the spacing effect on memory consolidation.

## Results

### PKCδ is required in MB neurons for LTM formation

To investigate the role of PKCδ in memory, we downregulated PKCδ expression at the adult stage in the MB neurons (see schematic in *Figure 1A*) and assessed memory formed using different protocols of aversive olfactory conditioning. These different protocols include the single association between an odor and shocks (1 x conditioning), which elicits a short-lived memory that will rapidly decay after a few hours; 5 spaced cycles of conditioning with 15 min of rest intervals (5 x spaced), which induces LTM; and 5 massed presentations without pauses (5 x massed), which leads to the formation of a less robust, cramming-like type of memory (*Bouzaiane et al., 2015*; *Heisenberg, 2003*; *Tully et al., 1994*). To spatially and temporally restrict PKCδ RNAi expression, we took advantage of the VT30559-Gal4 MB driver (*Plaçais et al., 2017*) combined with the ubiquitously expressed thermosensitive Gal4 inhibitor tub-Gal80ts (*McGuire et al., 2003*). This system allows inducing RNAi expression in the MBs of adult flies, by transferring them at 30 °C 2 days before conditioning. This protocol successfully decreased the whole-head PKCδ mRNA level (*Figure 1B*), thereby revealing PKCδ expression in MB neurons. When PKCδ was knocked down in adult MB neurons, flies subjected to 5 x spaced conditioning presented LTM impairment (*Figure 1C*). When flies of the same genotype were kept at 18 °C prior to conditioning, i.e. without induction of RNAi expression, LTM was normal (*Figure 1C*). In addition, the sensitivity to shocks and odors was normal in induced flies (*Table 1*). We then investigated if other types of aversive memory were also affected by PKCδ knockdown in adult MB neurons. No defect was detected when memory was measured 24 hr after 5 x massed conditioning (*Figure 1C*), and memory was not affected when tested 3 hr after single-cycle training (*Figure 1C*). All of these results were replicated with a second non-overlapping RNAi against PKCδ (*Figure 1—figure supplement 1A*), which downregulates PKCδ expression with similar efficiency (*Figure 1—figure supplement 1B*). Altogether, these series of experiments demonstrate that PKCδ is specifically required for LTM in adult MB neurons, fulfilling the first condition for a molecular effector of the spacing effect.

Next, we investigated whether PKCδ was activated in MB neurons following spaced training. We took advantage of the existence of a genetically encoded FRET-based fluorescent reporter of PKCδ-specific activity, δCKAR (*Kajimoto et al., 2010*; *Wu-Zhang et al., 2012*), and generated *Drosophila* lines carrying this sensor under UAS control. Expression of the δCKAR sensor in the adult MB neurons was achieved using the tub-Gal80ts;VT30559-Gal4 driver as for the behavior experiments. To assess the efficacy of the δCKAR sensor in *Drosophila* neurons, we first used two-photon in vivo imaging in the MB vertical lobes of naive flies to monitor the response of the δCKAR sensor to pharmacological activation of PKCδ, using PDBu (*Wu-Zhang et al., 2012*). PDBu application induced a robust response of the sensor as compared to solvent alone (*Figure 1D*). Because PDBu can activate other PKCs than PKCδ (*Alzamora et al., 2007*; *Wu-Zhang et al., 2012*), we ascertained the specificity of this response by performing the same experiment in flies expressing PKCδ RNAi in the MB at the adult stage, where PDBu failed to elicit a δCKAR response as compared to flies that do not carry PKCδ RNAi (*Figure 1E*). Notably, the order of magnitude of the maximum δCKAR response (2–3%) is consistent with what was previously measured in cellulo (*Wu-Zhang et al., 2012*).

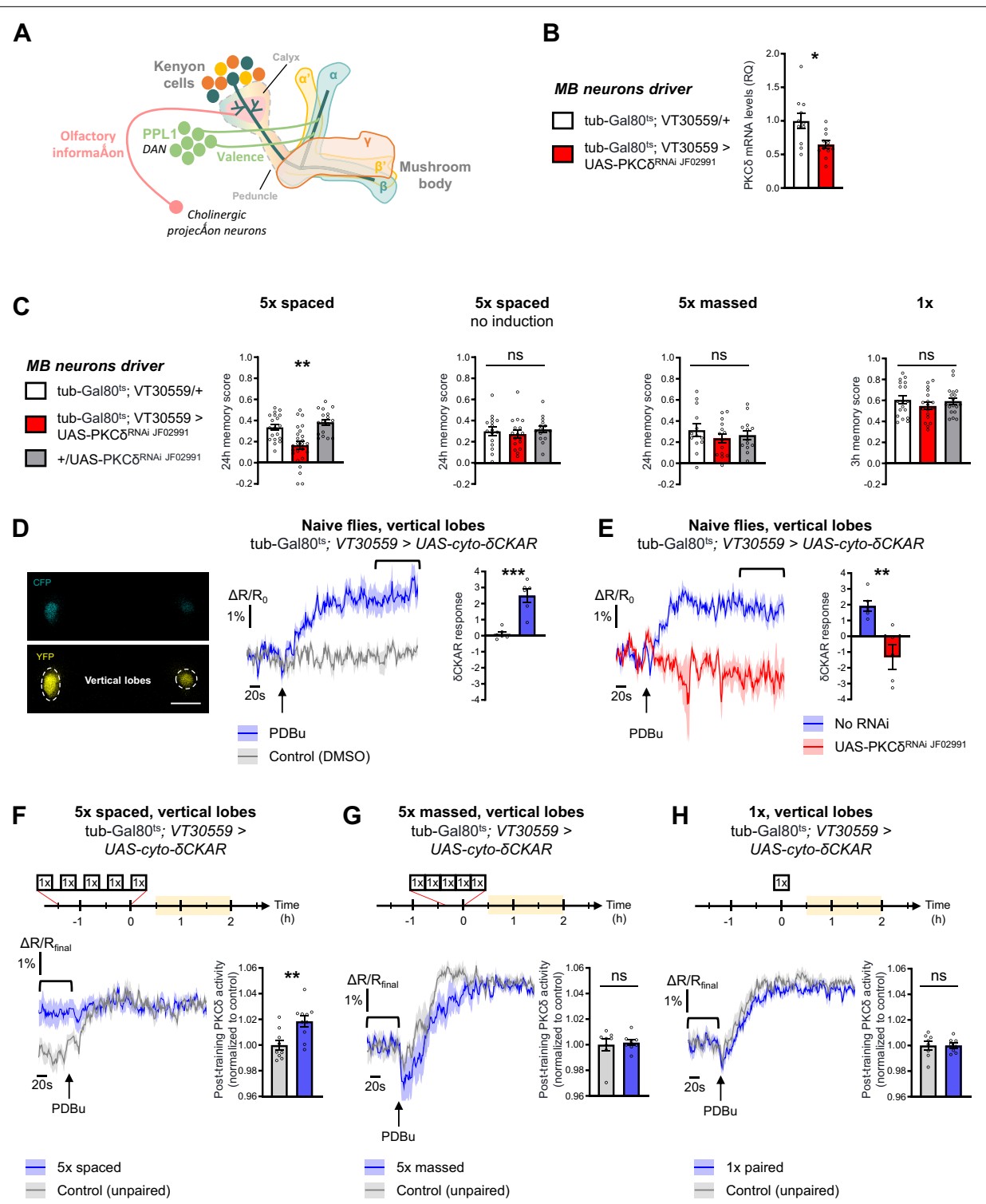

**Figure 1.** PKCδ is required in MB neurons for LTM formation. (**A**) Schematic representation of *Drosophila* MB. The MB includes ~2000 intrinsic neurons per brain hemisphere. Their cell bodies are located in the dorsal posterior part of the brain. MB neurons each send a single neurite into the neuropil, which first traverses the calyx, a dendritic region where MB neurons receive olfactory input from projections neurons, and then extends into a long axonal branch. The bundled axons of MB neurons form a fascicle called the peduncle, which traverses the brain to the anterior part, where axons branch to form medial and vertical lobes according to three major branching patterns – α/β, α'/β' and γ (*Aso et al., 2014*) – that define as many neuronal categories. The MB lobes receive input from dopaminergic neurons (the DANs), which signal stimuli of positive and negative valences in a region-specific manner (*Aso et al., 2014*). During associative learning, dopamine release on coincidentally odorant-activated MB output synapses modulates

*Figure 1 continued on next page*

*Figure 1 continued*

the synaptic drive to the network of MB output neurons, which bias subsequent odor-driven behavior (*Heisenberg, 2003*; *Hige, 2018*). Aversive LTM induced by spaced training is more specifically encoded within the α/β neurons (*Pascual and Préat, 2001*; *Séjourné et al., 2011*; *Yu et al., 2006*), and we previously showed that LTM retrieval involves the depression of an attraction-mediated pathway efferent from the MB vertical lobes (*Aso et al., 2014*; *Bouzaiane et al., 2015*; *Dolan et al., 2018*; *Séjourné et al., 2011*). However, according to another recent study, LTM retrieval mobilizes in parallel another MB output circuit efferent from the medial lobes (*Jacob and Waddell, 2020*). (**B**) Expression of PKCδ RNAi in adult MB neurons induced a significant reduction in the mRNA level of PKCδ measured by RT-qPCR in fly heads. Relative Quantification (RQ) was performed, indicating the foldchange of mRNA levels relative to the control genotype (n=11, $t_{20}$=2.83, p=0.010). (**C**) PKCδ knockdown in adult MB neurons impaired memory after 5 x spaced conditioning (n=17–25, $F_{2,58}$=12.59, p<0.0001). Without the induction of PKCδ RNAi expression, memory formed after 5 x spaced conditioning was normal (n=15–17, $F_{2,45}$=0.41, p=0.67). Memory formed after 5 x massed training (n=13–14, $F_{2,37}$=0.65, p=0.53) and 1 x training (n=18, $F_{2,51}$=0.81, p=0.45) was normal in flies knocked down for PKCδ in adult MB neurons. (**D**) The cyto-δCKAR sensor was expressed in adult MB neurons and visualized in the CFP and YFP channels. Cytosolic PKCδ activity levels are recorded within the vertical lobes of the MBs (indicated with dashed line). Scale bar = 50 µm (valid for both channels). In naive flies, application of 250 µM of PDBu (black arrow), a pharmacological activator of PKCδ, resulted in the increase of the cyto-δCKAR response, reaching a plateau, as compared to the DMSO control (n=6, $t_{10}$=5.66, p=0.0002). Quantification of the mean cyto-δCKAR response was performed 280 s after PDBu application on a time window of 560 s (black line). (**E**) In naive flies, application of 250 µM of PDBu (black arrow) resulted in an increase in the cyto-δCKAR response that is abolished when PKCδ is knocked down in adult MB neurons (n=5–6, $t_9$=4.18, p=0.0024). Quantification of the mean cyto-δCKAR response was performed 280 s after PDBu application on a time window of 560 s (black line). (**F**) To compare post-conditioning cytosolic PKCδ activities (between 30 min and 2 hr post-conditioning, in yellow on the imaging time frame), cyto-δCKAR traces were normalized to the plateau value reached after addition of PDBu (saturation of the sensor), thus the activity level of cytosolic PKCδ is estimated as the cyto-δCKAR signal value before PDBu application. Cytosolic PKCδ activity is increased in the vertical lobes after 5 x spaced associative paired conditioning as compared to a non-associative spaced conditioning (unpaired) protocol (n=9–10, $t_{17}$=3.18, p=0.0055). Quantification of the mean post-training PKCδ activity was performed on a time window of 120 s before PDBu application (black line). (**G**) After 5 x massed paired conditioning, cytosolic PKCδ activity was not changed as compared to 5 x massed unpaired conditioning (n=8, $t_{14}$=0.33, p=0.75). (**H**) Similarly, after 1 x paired conditioning, cytosolic PKCδ activity was not changed as compared to 1 x unpaired conditioning (n=8, $t_{14}$=0.0041, p=0.99). Data are expressed as mean ± SEM with dots as individual values, and were analyzed by one-way ANOVA with post hoc testing by the Tukey pairwise comparisons test (**C**) or by unpaired two-sided t-test (**B, D–H**). Asterisks refer to the least significant p-value of post hoc comparison between the genotype of interest and the genotypic controls (**C**), or to the p-value of the unpaired t-test comparison (**B, D–H**) using the following nomenclature: *p<0.05, **p<0.01, ***p<0.001, ns: not significant, p>0.05. See also *Figure 1—figure supplement 1* and *Table 1*.

The online version of this article includes the following source data and figure supplement(s) for figure 1:

**Source data 1.** Source data displayed on *Figure 1*.

**Figure supplement 1.** Control experiments for behavior analysis and δCKAR imaging experiments.

**Figure supplement 1—source data 1.** Source data displayed on *Figure 1—figure supplement 1*.

**Table 1.** Sensory acuity controls in PKC δ knockdown flies.
Control experiments for olfactory acuity and electric shock avoidance: the expression of either of the two PKC δ RNAi used in this study, in MB neurons at the adult stage had no significant effect on olfactory acuity, or on the avoidance of electric shocks.

| | | | Naive odor avoidance | | | |
| | Shock avoidance | | Octanol | | Methylcyclohexanol | |
| Genotypes | Mean ± s.e.m. | Statistics | Mean ± s.e.m. | Statistics | Mean ± s.e.m. | Statistics |
|---|---|---|---|---|---|---|
| tubGal80ts;VT30559/+ | 0.63±0.057 | | 0.74±0.053 | | 0.72±0.069 | |
| tubGal80ts;VT30559 >UAS-PKCδRNAi JF02991 | 0.76±0.033 | $F_{2,33}$=2.60; p=0.09; n=12 | 0.53±0.070 | $F_{2,27}$=5.60; p=0.0092*; n=9–11 | 0.60±0.082 | $F_{2,27}$=1.27; p=0.30; n=9–11 |
| UAS-PKCδRNAi JF02991/+ | 0.64±0.039 | | 0.44±0.073 | | 0.56±0.069 | |
| tubGal80ts;VT30559/+ | 0.47±0.052 | | 0.63±0.040 | | 0.66±0.046 | |
| tubGal80ts;VT30559 >UAS-PKCδRNAi KK109117 | 0.42±0.039 | $F_{2,33}$=0.38; p=0.69; n=12 | 0.70±0.033 | $F_{2,33}$=2.76; p=0.078; n=12 | 0.67±0.034 | $F_{2,33}$=0.64; p=0.53; n=12 |
| UAS-PKCδRNAi KK109117/+ | 0.43±0.047 | | 0.57±0.047 | | 0.61±0.050 | |
| tubGal80ts,c739/+ | 0.45±0.060 | | 0.58±0.050 | | 0.46±0.044 | |
| tubGal80ts,c739 >UAS-PKCδRNAi JF02991 | 0.40±0.053 | $F_{2,39}$=0.61; p=0.55; n=14 | 0.45±0.036 | $F_{2,33}$=2.48; p=0.099; n=12 | 0.47±0.051 | $F_{2,33}$=0.94; p=0.40; n=12 |
| UAS-PKCδRNAi JF02991/+ | 0.49±0.059 | | 0.48±0.045 | | 0.54±0.048 | |

[*]Tukey post hoc comparison between the genotype of interest and controls are not significant: tubGal80ts;VT30559 >UAS-PKCδRNAi JF02991 vs tubGal80ts;VT30559/+: ns tubGal80ts;VT30559 >UAS-PKCδRNAi JF02991 vs UAS-PKCδRNAi JF02991/+: ns tubGal80ts;VT30559/+vs UAS-PKCδRNAi JF02991/+: **.

The online version of this article includes the following source data for table 1:

**Source data 1.** Source data displayed on *Table 1*.

Having established the responsiveness and the specificity of the δCKAR sensor, we sought to use this tool to measure the PKCδ activity level after conditioning in the MB vertical lobes, which are privileged sites of LTM encoding (*Pascual and Préat, 2001*; *Séjourné et al., 2011*; *Yu et al., 2006*). To compare post-conditioning PKCδ activity between different protocols, δCKAR traces were normalized to the plateau value reached after saturation of the sensor by addition of PDBu, so that the activity level of PKCδ in each individual fly could be estimated as a δCKAR signal value before PDBu application (*Figure 1F*). Measurements were performed in a time window of 30 min to 2 hr after the end of training, when increased mitochondrial metabolism in MB vertical lobes after 5 x spaced was previously reported (*Plaçais et al., 2017*; *Rabah et al., 2023*). Using this procedure, we observed that the PKCδ activity level was increased following spaced training, as compared to control flies that were submitted to a non-associative unpaired spaced protocol (*Figure 1F*). Importantly, the PKC substrate-uncompetitive inhibitor Bis IV (*Wu-Zhang et al., 2012*; *Figure 1—figure supplement 1C*) was still able to induce a decrease in PKCδ activity after spaced training (*Figure 1—figure supplement 1D*), ruling out the alternative hypothesis that lack of a PDBu-induced δCKAR response following spaced training might stem from an inhibition of PKCδ expression or a decrease of its activity below the detection threshold of the δCKAR sensor. Therefore, we conclude from our results that 5 x spaced

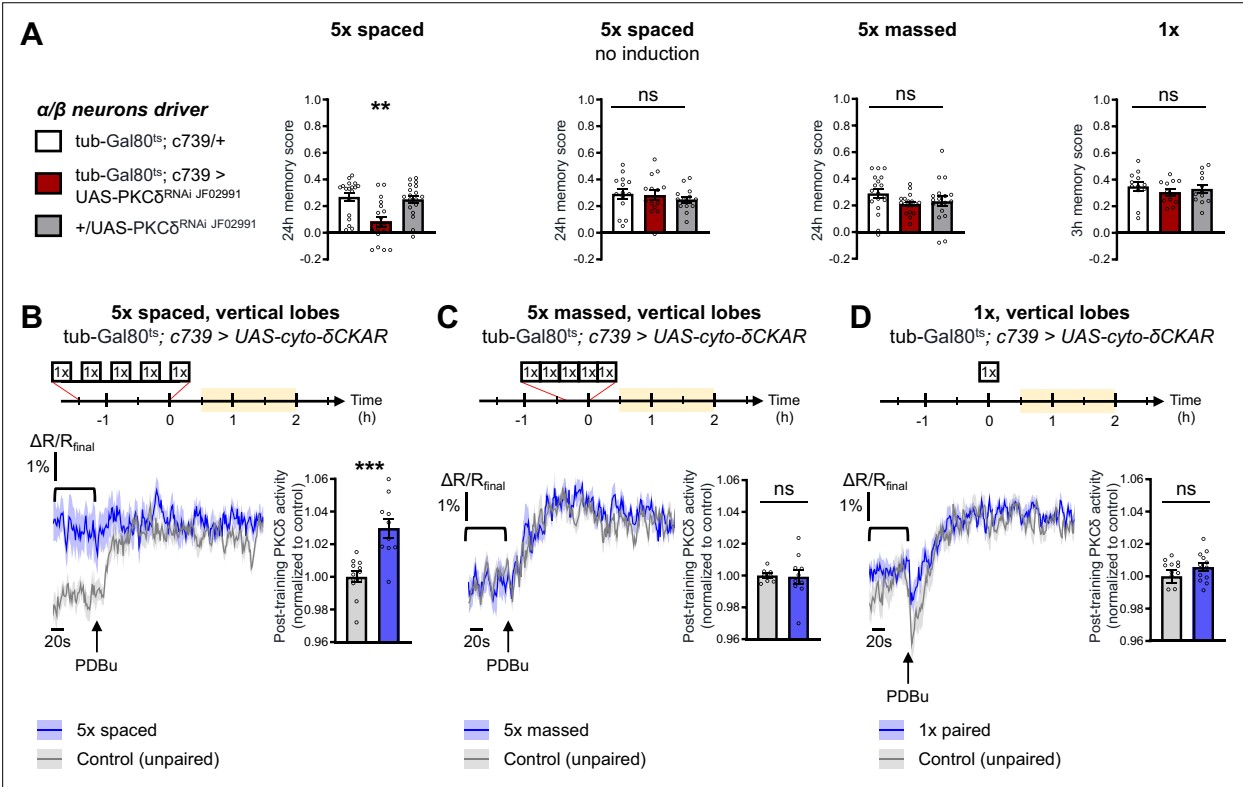

**Figure 2.** PKCδ is required in α/β neurons for LTM formation. (**A**) When PKCδ was knocked down specifically in the α/β neurons of MBs, memory after 5 x spaced conditioning was impaired (n=18, $F_{2,51}$=9.52, p=0.0003. Without PKCδ RNAi induction, memory after 5 x spaced conditioning was normal (n=14, $F_{2,39}$=0.47, p=0.63). Memory was normal after 5 x massed training (n=18, $F_{2,51}$=1.771, p=0.1805) as well as after 1 x training (n=12, $F_{2,33}$=0.51, p=0.61) in flies expressing PKCδ RNAi in adult α/β MB neurons. (**B**) Using the same approach as detailed in *Figure 1F*, the post-training activity of cytosolic PKCδ was measured specifically in the α/β neurons, between 30 min and 2 hr post-conditioning (in yellow on the imaging time frame). After 5 x spaced paired conditioning, cytosolic PKCδ activity was increased in the α lobes as compared to unpaired conditioning (n=10–12, $t_{20}$=4.58, p=0.0002). (**C**) Following 5 x massed conditioning, cytosolic PKCδ activity in the α lobes was not changed compared to unpaired conditioning (n=8–10, $t_{16}$=0.13, p=0.90). (**D**) After 1 x paired conditioning, cytosolic PKCδ activity in the α lobes also remained unchanged compared to unpaired conditioning (n=12, $t_{22}$=1.16, p=0.26). Data are expressed as mean ± SEM with dots as individual values, and were analyzed by one-way ANOVA with post hoc testing by the Tukey pairwise comparisons test (**A**) or by unpaired two-sided t-test (**B–D**). Asterisks refer to the least significant P-value of a post hoc comparison between the genotype of interest and the genotypic controls (**A**) or to the p-value of the unpaired t-test comparison (**B–D**) using the following nomenclature: **p<0.01, p***<0.001, ns: not significant, p>0.05. See also *Table 1*.

The online version of this article includes the following source data for figure 2:

**Source data 1.** Source data displayed on *Figure 2*.

conditioning actually induced a marked enhancement of PKCδ activity in MB neurons. In addition, neither massed nor 1 x training elicited such an increase in PKCδ activity levels (*Figure 1G–H*), which supports the specific effect of spaced conditioning on PKCδ activation and is consistent with the outcome of our behavioral experiments. Of note, expression of the cyto-δCKAR sensor (or of any of the other imaging probes used in this study) in the MB neurons at adult stage did not impact the formation of memory upon the various training protocols employed here (5 x spaced, 5 x massed and 1 x, *Figure 1—figure supplement 1E*).

The specific requirement of PKCδ in LTM formation prompted us to investigate whether it was more particularly required in α/β neurons of the MB, one of the three anatomical categories of MB neurons and a subpopulation known to be pivotal in LTM encoding (*Pascual and Préat, 2001*; *Séjourné et al., 2011*; *Yu et al., 2006*). We therefore restricted PKCδ RNAi expression to the α/β neurons exclusively at the adult stage, by means of the tub-Gal80^ts;c739-Gal4 inducible driver (*Aso et al., 2009*). This strongly impaired LTM performance after spaced training (*Figure 2A*). When RNAi expression was not induced, LTM was normal (*Figure 2A*). Naive odor and shock avoidance were unaffected by PKCδ knockdown in adult α/β neurons (*Table 1*). Neither memory after massed training nor 3 hr memory after 1 x training were impaired (*Figure 2A*). Hence, we show that PKCδ is more specifically required in α/β neurons for LTM formation. We then investigated PKCδ activity after spaced conditioning more specifically in α/β MB neurons using the δCKAR sensor. Consistent with the requirement of PKCδ specifically for LTM in the α/β neurons, we observed an increase in PKCδ post-training activity after spaced conditioning (*Figure 2B*), an activation that was not elicited by massed training (*Figure 2C*) nor by a single-cycle of training (*Figure 2D*).

Altogether, we demonstrate here that PKCδ is specifically activated after spaced training and is required for LTM formation in MB α/β neurons, placing it as a key player of the spacing effect.

## PKCδ regulates mitochondrial pyruvate metabolism for LTM

Increased mitochondrial metabolic activity in MB neurons after spaced training is known to be critical for the initiation of LTM formation (*Plaçais et al., 2017*). The level of mitochondrial pyruvate metabolism in MB neurons can be modulated thanks to the pyruvate dehydrogenase (PDH) complex (*Plaçais et al., 2017*). Indeed, the activity of PDH is regulated by its level of phosphorylation: while the PDH kinase (PDK) inhibits PDH, the PDH phosphatase (PDP) activates it (*Lavington et al., 2014*; *Figure 3A*). To further investigate whether the upregulation of the pyruvate flux in itself initiates LTM formation, we expressed exclusively at the adult stage an RNAi against PDK in MB neurons, which increases the pyruvate uptake rate by mitochondria in the MB neurons (*Plaçais et al., 2017*). The PDK RNAi efficiently downregulated PDK expression in neurons (*Figure 3—figure supplement 1A*). Upon PDK knock-down in the adult MB neurons, we observed that only a single-cycle of associative conditioning was sufficient to form LTM, measured 24 hr after conditioning, while genotypic control flies fail to remember at that timepoint (*Figure 3A*). Normal LTM formation upon 5 x spaced training, as well as normal odor and shock avoidance were previously confirmed, and a facilitation of LTM was previously reported following two cycles of spaced training (*Plaçais et al., 2017*). Altogether, this result, as well as our previous report (*Plaçais et al., 2017*), show that the activity level of PDK, and the resulting regulation of PDH, determines the number of training repeats required to form LTM.

We have shown that PKCδ activation occurs in the first hours after spaced training, that is concomitantly to mitochondrial metabolic activation. As PKCδ can modulate mitochondrial metabolism in other tissues (*Acin-Perez et al., 2010*; *Kim and Hammerling, 2020*), we wondered whether PKCδ activation in neurons was involved in this process. To address this question, we first asked whether pharmacological PKCδ activation using PDBu in a naive context could be sufficient to upregulate the mitochondrial pyruvate flux in MB neurons. To this end, we used in vivo two-photon imaging of the genetically encoded pyruvate sensor Pyronic (*San Martín et al., 2014*) expressed in all MB neurons. We employed a previously characterized protocol (*Plaçais et al., 2017*) to obtain a dynamic measurement of the pyruvate flux to mitochondria, by measuring the slope of pyruvate accumulation following the injection of sodium azide, a potent inhibitor of the mitochondrial respiratory chain (complex IV) (*Figure 3—figure supplement 1B*). In the brains of naive flies, activation of PKCδ through PDBu application resulted in an upregulated pyruvate flux to the mitochondria of the MB vertical lobes (*Figure 3—figure supplement 1C*). This upregulation did not occur when PKCδ was knocked down in adult MB neurons (*Figure 3—figure supplement 1C*), indicating that the increased

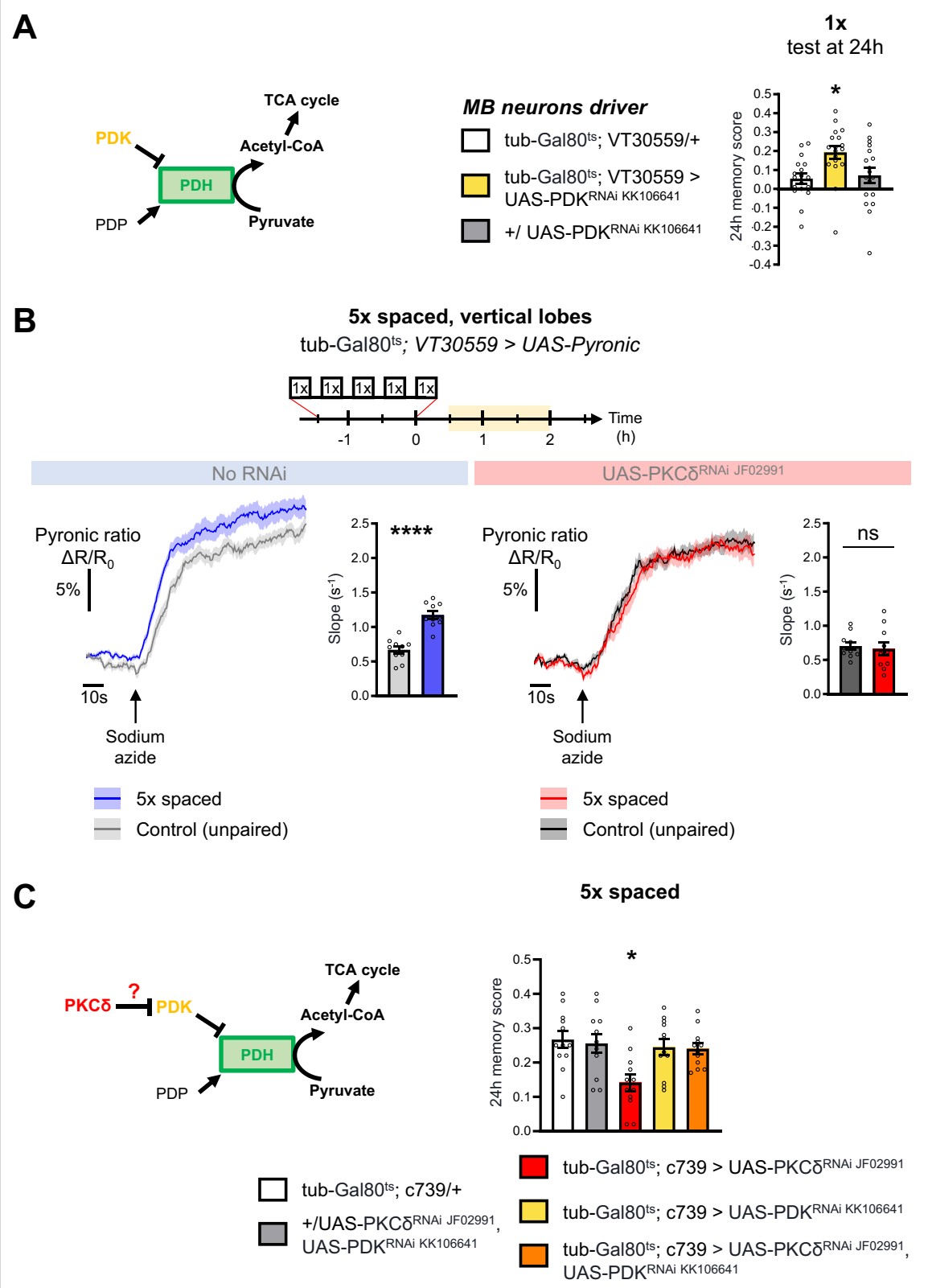

**Figure 3.** PKCδ regulates mitochondrial pyruvate metabolism upon LTM formation. (**A**) Left panel: schema of the regulation of the pyruvate dehydrogenase (PDH) complex. In mitochondria, PDHC catalyzes the conversion of pyruvate to acetyl-CoA, which enters the tricarboxylic acid cycle (TCA). PDH can be inactivated through phosphorylation by PDK. In contrast, PDP can activate PDH via its dephosphorylation. Right panel: flies expressing an RNAi against PDK in MB neurons exclusively at the adult stage showed increased memory measured at 24 hr after a single cycle of

*Figure 3 continued on next page*

*Figure 3 continued*

training as compared to their genotypic controls (n=18, $F_{2,51}$=5.09, p=0.0097). (**B**) The pyruvate sensor Pyronic was expressed in adult MB neurons and the pyruvate FRET signal was quantified in the vertical lobes. In control flies, spaced training elicited a faster pyruvate accumulation in axons of MB neurons after sodium azide application (5 mM; black arrow) as compared to non-associative unpaired training (left panel, slope measurement n=10, $t_{18}$=6.751, p<0.0001). PKCδ knockdown in adult MB neurons impaired the spaced training induced increase in pyruvate accumulation in axons of MB neurons following sodium azide application (right panel, slope measurement n=10, $t_{18}$=0.38, p=0.71). As for **Figures 1 and 2**, imaging was performed between 30 min and 2 hr post-conditioning, represented in yellow on the imaging time frame. (**C**) Left panel: schema of our model, asking whether PKCδ intervene upstream of the PDH complex. Here, we show that PKCδ regulates PDH activity via PDK inhibition. Right panel: after 5 x spaced conditioning, flies coexpressing the PDK and PKCδ RNAi in the α/β MB neurons at the adult stage exhibited normal memory formation as compared to genotypic controls. Flies solely expressing the PKCδ RNAi in α/β neurons at the adult stage exhibited the reported LTM defect, whereas flies that were only knocked down for PDK in adult α/β neurons formed normal memory (n=12, $F_{4,55}$=4.75, p=0.0023). Data are expressed as mean ± SEM with dots as individual values, and were analyzed by unpaired two-sided t-test (**B**) or by one-way ANOVA with post hoc testing by the Tukey pairwise comparisons test (**A,C**). Asterisks refer to the p-value of the unpaired t-test comparison or to the least significant p-value of post hoc comparison between the genotype of interest and the genotypic controls using the following nomenclature: *p<0.05, ****p<0.0001, ns: not significant, p>0.05. See also *Figure 3—figure supplement 1*.

The online version of this article includes the following source data and figure supplement(s) for figure 3:

**Source data 1.** Source data displayed on *Figure 3*.

**Figure supplement 1.** PKCδ activation by PDBu application increases the MB neuronal pyruvate flux.

**Figure supplement 1—source data 1.** Source data displayed on *Figure 3—figure supplement 1*.

pyruvate consumption relies on PKCδ activation by PDBu. These results reveal that in naive flies, PKCδ activation is sufficient to increase the pyruvate flux to mitochondria in MB neurons.

We then investigated whether PKCδ mediates metabolic upregulation after spaced training. Expression of the Pyronic probe in the MB neurons at adult stage did not compromise the formation of memory after 5 x spaced conditioning (neither after 5 x massed nor 1 x conditioning) (*Figure 1—figure supplement 1E*), making this system suitable for the study of conditioned flies. As previously reported (*Plaçais et al., 2017*), we observed an increased pyruvate flux in MB vertical lobes following spaced conditioning (*Figure 3B*) during the same time frame as for PKCδ activation. Remarkably, this effect was abolished by PKCδ knockdown in adult MB (*Figure 3B*). Therefore, we conclude that a critical role of PKCδ in LTM formation is to upregulate mitochondrial pyruvate metabolism for LTM gating.

Next, we examined how PKCδ could regulate mitochondrial pyruvate consumption. Given that the genetic inhibition of PDK facilitates LTM formation (*Figure 3A*), and that, in non-neuronal cells, PKCδ has been described as an inhibitor of PDK (*Acin-Perez et al., 2010*; *Kim and Hammerling, 2020*), we hypothesized that a similar mechanism – that is PDK inhibition resulting in PDH activation – may occur in neurons in the context of LTM formation. To test this hypothesis, we reasoned that genetic inhibition of PDK may alleviate the LTM defect elicited by PKCδ knockdown (as described in *Figure 1C*). Indeed, we observed that coexpression of PDK and PKCδ RNAis in adult α/β MB neurons did not induce a memory impairment, whereas expression of the PKCδ RNAi alone induced an LTM defect as expected (*Figure 3C*). Thus, inhibiting PDK genetically is sufficient to fully rescue the memory defect induced by loss of PKCδ, showing that upregulating pyruvate flux via the PDH complex is the critical function of PKCδ in LTM formation.

## MP1 neurons activate PKCδ in the mitochondria of MB neurons via DAMB signaling

As detailed in the introduction, the metabolic upregulation induced by spaced training has been shown to be triggered by early post-learning activity of MP1 dopamine neurons, through the Gq-coupled DAMB receptor (*Plaçais et al., 2017*). We therefore asked if MP1 neuronal dopamine signaling through the DAMB receptor could activate PKCδ and induce its mitochondrial translocation, which would be detected as an increase

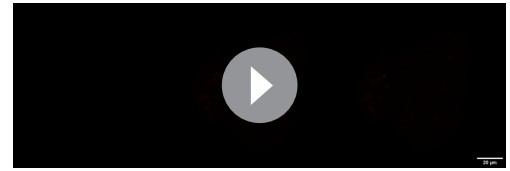

**Video 1.** Video of mito-δCKAR and mito-DsRed colocalization z-stack. Video of the full z-stack composed of 28 images showing the entire MB soma of the representative fly shown on *Figure 4—figure supplement 1B*. The YFP channel is on the left, DsRed is in the middle, and the merged channels are shown on the right. Scale bar = 20 μm.
https://elifesciences.org/articles/92085/figures#video1

in PKCδ activity specifically at the level of the mitochondria. For this, we expressed a mitochondria-addressed δCKAR sensor, mito-δCKAR (*Figure 4—figure supplement 1* and *Video 1*), in MB neurons, and monitored PKCδ activity while artificially activating MP1 neurons in naive flies using dTrpA1, a heat-sensitive cation channel that allows to induce neuronal firing through acute temperature increase (*Hamada et al., 2008*; *Plaçais et al., 2012*). At the level of the peduncle, where MP1 neurons project onto the α/β MB neurons (*Aso et al., 2014*; *Figure 4A*), flies subjected to the activation of MP1 neurons showed an increased PKCδ mitochondrial activity as compared to flies that received a similar thermal treatment but did not express dTrpA1 (*Figure 4B*). Strikingly, in flies that were additionally knocked down for the DAMB receptor in the MBs, MP1 neuron activation failed to increase PKCδ mitochondrial activity (*Figure 4B*). Interestingly, we found that PKCδ mitochondrial activity could also be increased in the vertical lobes of the MB in response to MP1 activation (*Figure 4C*), although to a lesser extent compared to the peduncle region (*Figure 4B*). This mitochondrial activation of PKCδ in the vertical lobes also depends on DAMB, as its knock-down in the MBs hindered PKCδ activity level increase (*Figure 4C*). Altogether, these data show that DAMB signaling from MP1 neurons triggers increased PKCδ mitochondrial activity, reflecting its translocation to mitochondria in MB neurons.

To explore whether PKCδ mitochondrial activation and translocation occur upon 5 x spaced conditioning for LTM formation, we expressed the mito-δCKAR sensor in α/β neurons and measured PKCδ activity after 5 x spaced training, using the same method as in *Figure 1F–H*. Such expression of mito-δCKAR in the MB neurons at adult stage did not affect the formation of memory upon 5 x spaced conditioning (neither after 5 x massed nor 1 x conditioning; *Figure 1—figure supplement 1E*). At the level of the peduncle, we found that 5 x spaced training elicited a marked increase in PKCδ mitochondrial activity, as compared to the corresponding unpaired protocol (*Figure 5A*). Remarkably, DAMB knockdown in adult α/β neurons hampered PKCδ activation in mitochondria after 5 x spaced training (*Figure 5A*). We observed a similar DAMB-dependent effect at the level of the α lobe (*Figure 5B*), as well as in the β lobe (*Figure 5—figure supplement 1A*).

As the post-learning activation of MP1 neurons is known to last up to 2 hr after the last cycle of 5 x spaced conditioning (*Plaçais et al., 2012*), we then asked whether PKCδ activated state in the α/β neurons' mitochondria could be maintained beyond 2 hr. We therefore measured PKCδ activity between 3 hr and 4 hr 30 min after 5 x spaced conditioning and found that PKCδ mitochondrial activation was still occurring at that time point, both at the level of the peduncles (*Figure 5C*) and of the α lobes (*Figure 5E*). However, at 8 hr post-conditioning, PKCδ activity was back to its baseline level in the peduncles (*Figure 5D*) and α lobes (*Figure 5F*). Altogether, our data show that MP1-DAMB signaling induced by spaced conditioning results in PKCδ sustained activation at the level of the mitochondria, where it upregulates the PDH complex to gate LTM formation through increased pyruvate flux. PKCδ remains activated in mitochondria for more than 3 hr, and it returned to its unactivated state by 8 hr post-training.

## Spaced training prolongs learning-induced metabolic enhancement

The results obtained thus far uncover a DAMB/PKCδ cascade that mediates an increase in pyruvate consumption by MB neuronal mitochondria upon 5 x spaced training for LTM formation. Intriguingly, single-cycle training also induces a metabolic upregulation of similar magnitude, typically observed 1–2 hr after training, at the level of the MB neurons' somas (as previously reported in *Rabah et al., 2023*), vertical lobes (*Figure 6A* and *Rabah et al., 2023*) and medial lobes (*Figure 6—figure supplement 1C*). However, we observed that the metabolic enhancement induced by single-cycle training in those compartments faded away rapidly, as it was no longer detectable 3 hr after 1 x training (vertical lobes *Figure 6B*, somas *Figure 6—figure supplement 1A* and medial lobes *Figure 6—figure supplement 1D*). Moreover, the metabolic upregulation occurring after 1 x training was not dependent on DAMB signaling (*Figure 6C*), nor did it require PKCδ activity in MB neurons (*Figure 6D*), contrary to what was observed after spaced training (see *Plaçais et al., 2017* for DAMB and *Figure 3B* for PKCδ). This led us to hypothesize that the role of this DAMB/PKCδ cascade after spaced training is to extend the duration of the metabolic enhancement in MB neurons. Indeed, measurements performed 3–4 hr as well as 8–9 hr after the last cycle of spaced training revealed that increased metabolic activity was still occurring at the level of the vertical lobes (*Figure 6E–F*). In contrast, measurements performed 24 hr after spaced training showed that the metabolic enhancement had stopped by that time point (*Figure 6G*). In the medial lobes, the observed increased pyruvate flux after 5 x spaced

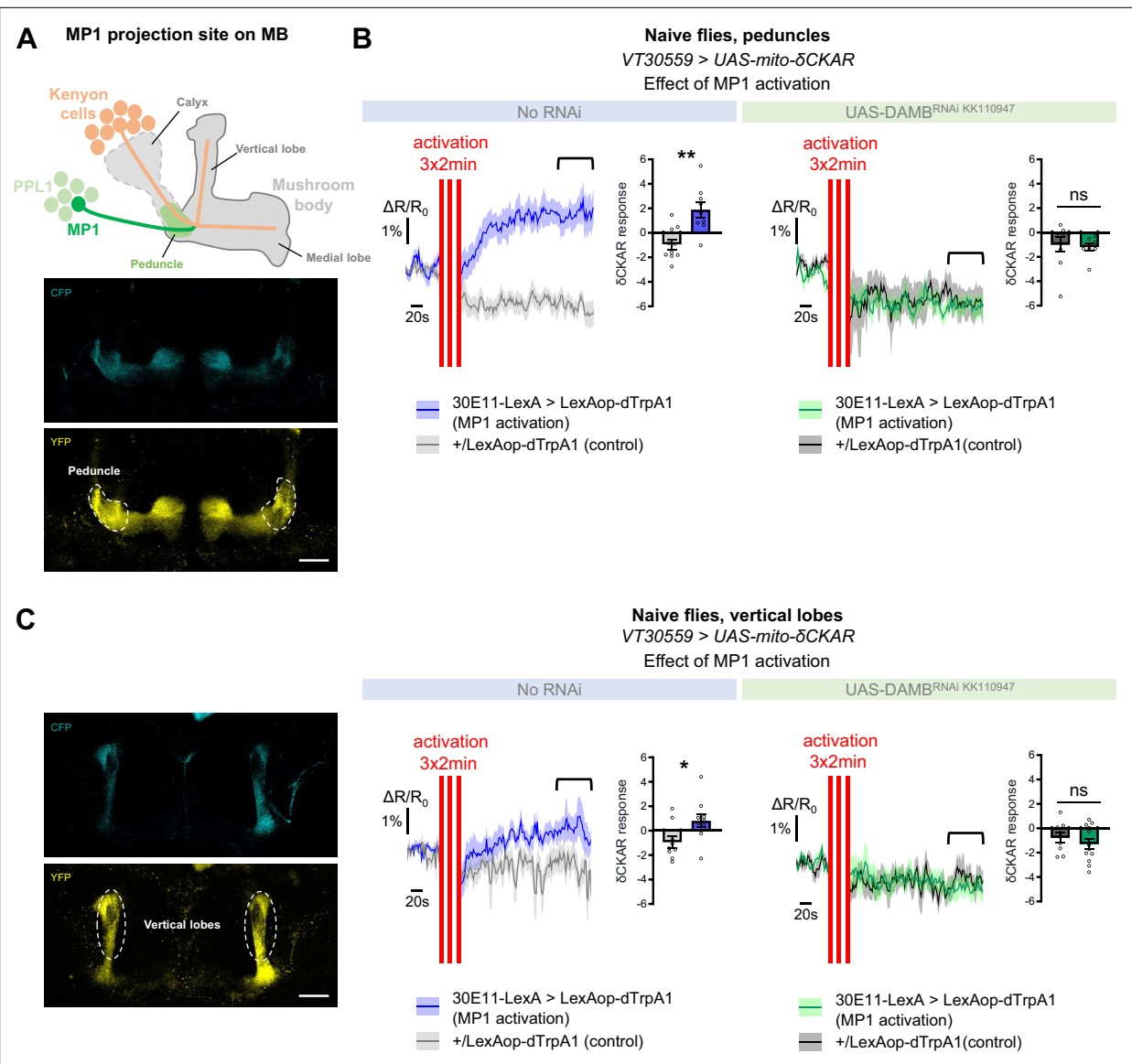

**Figure 4.** MP1 neurons control PKCδ activity in the MB neurons via DAMB signaling. (**A**) Top panel: Schema of the MB structure and its afferent MP1 neuron. The dopaminergic neuron MP1 (depicted in green) in the PPL1 cluster projects onto the peduncle area of the MBs. Lower panel: The mito-δCKAR sensor was expressed in adult MB neurons and visualized in the CFP and YFP channels. The region of recording for panel B is at the level of the MB peduncles (dashed line). Scale bar = 30 μm (valid for both channels). (**B**) Naive flies expressing the mito-δCKAR sensor in MB neurons together with the dTrpA1 (a heat-sensitive cation channel) in MP1 neurons (30E11-LexA driver) were subjected to a thermal treatment consisting of three 2 min periods at 30 °C separated by 2 min (red vertical lines); control flies expressed the mito-δCKAR in MB neurons but not the dTrpA1 channel (no 30E11-LexA driver). Mitochondrial PKCδ activity was recorded before (baseline) and immediately after the activation periods. Quantification of the mean mito-δCKAR response was performed 120 s after the last cycle of thermal activation on a time window of 480 s (black line). In naive flies expressing the dTrpA1 channel in MP1 neurons, activation of MP1 increased mitochondrial PKCδ activity as compared to control flies (n=9–10, $t_{17}$=3.83, p=0.0013). When DAMB was knocked down in MB neurons, MP1 activation failed to increase mitochondrial PKCδ activity as compared to control flies (n=8–9, $t_{15}$=0.31, p=0.76). (**C**) The mito-δCKAR sensor was expressed in adult MB neurons and mitochondrial PKCδ activity was recorded in the vertical lobes (dashed line). As in the peduncle region, in naive flies expressing the dTrpA1 channel in MP1 neurons activation of MP1 increased mitochondrial PKCδ activity in the vertical lobes as compared to control flies (n=9–10, $t_{17}$=2.37, p=0.030). DAMB knock-down in the MB neurons also prevented any increase of PKCδ mitochondrial activity in the vertical lobes upon the thermogenic activation of MP1 neurons, as compared to the genotypic control (n=9–12, $t_{19}$=0.83, p=0.42). Data are expressed as mean ± SEM with dots as individual values and were analyzed by unpaired two-sided t-test. Asterisks refer to the p-value of the unpaired t-test comparison using the following nomenclature: *p<0.05, **p<0.01, ns: not significant, p>0.05. Scale bar = 30 μm (valid for both channels). See also *Figure 4—figure supplement 1* and *Video 1*.

The online version of this article includes the following source data and figure supplement(s) for figure 4:

**Source data 1.** Source data displayed on *Figure 4*.

*Figure 4 continued on next page*

*Figure 4 continued*

**Figure supplement 1.** Subcellular addressing of the cyto-δCKAR and mito-δCKAR probes.

training (*Figure 6—figure supplement 1E*) also persisted for 8–9 hr post-conditioning (*Figure 6—figure supplement 1F*), whereas in the somas, the reported upregulation of mitochondrial pyruvate uptake (*Pavlowsky et al., 2024*) was not sustained and faded away rapidly at 3 hr after 5 x spaced conditioning (*Figure 6—figure supplement 1B*). Altogether, our results demonstrate that spaced training specifically recruits dopamine/DAMB/PKCδ signaling to perpetuate the metabolic activation in MB neurons' axonal compartment, which is critical for initiating LTM formation (*Figure 7*).

## Discussion

In this study, we unveiled a detailed molecular mechanism underlying the spacing effect on memory consolidation. We established that PKCδ, a known regulator of mitochondrial pyruvate metabolism in several peripheral tissues, is a critical player in LTM formation in α/β MB neurons, while being dispensable for less stable forms of aversive memory. Additionally, we imported in *Drosophila* a genetically encoded FRET sensor for PKCδ (*Kajimoto et al., 2010*; *Wu-Zhang et al., 2012*), which allowed observing the PKCδ activity level in vivo. Thus, we reported a specific activation and mitochondrial translocation of PKCδ following spaced training, but not 1 x or massed training. We established that specific dopaminergic neurons (MP1 neurons), which are required early after spaced training for LTM formation (*Plaçais et al., 2017*), activate PKCδ translocation to mitochondria, through a specific Gq-coupled dopamine receptor, DAMB. Consistent with the role of this dopamine signaling that we previously reported (*Plaçais et al., 2017*), we showed here that activated PKCδ persistently upregulates pyruvate metabolism, thereby allowing LTM formation. Altogether, our data demonstrate that PKCδ, acting as an activator of mitochondrial metabolism, is a neuronal gatekeeper of LTM formation. More generally, our findings provide a detailed mechanistic description of how dopamine signaling orchestrates memory consolidation through sustained modulation of neuronal energy fluxes, thereby resolving within a single gating mechanism the cognitive and metabolic constraints linked to LTM formation.

It is particularly striking that local dopaminergic signaling specifically delivered at the level of the MBs peduncle is able to globally activate a kinase in the various axonal compartments of the MBs. This observation calls for the understanding of the transfer mechanisms taking place to propagate PKCδ activation and translocation to mitochondria from the peduncle to the vertical and medial lobes. Given that MBs distinct modules are not separated by physical barriers, and that DAMB/PKCδ activation lasts several hours, two hypotheses can be proposed to explain this phenomenon. First, passive diffusion of activated PKCδ from the peduncle disseminating in the MBs axons until it translocates to mitochondria located in the vertical and medial lobes. Kinases are indeed known to be able to passively diffuse, an important mechanism for signal transduction (*Kazmierczak and Lipniacki, 2009*). Second, we recently found that mitochondrial motility is increased in the first hours following spaced training (*Pavlowsky et al., 2024*): mitochondria move from the MB neurons' somas along the axons to the lobes in order to sustain the increased energy demand for the formation of LTM. One can thus hypothesize that the signal diffusion could also occur via the motility of mitochondria – upon spaced conditioning, activated PKCδ could 'hitchhike' into mitochondria at the level of the peduncle, and these PKCδ-loaded, activated mitochondria would then further move to the lobes. These two hypotheses are not mutually exclusive, and are both compatible with the interesting observation that upon MP1 activation, the measured level of PKCδ mitochondrial activation is stronger in the peduncle (*Figure 4B*) than in the vertical lobes (*Figure 4C*).

Although many different enzymes of the PKC family have been implicated in learning and memory in a large variety of model organisms (*Van Der Zee and Douma, 1997*), the particular contribution of PKCδ in memory formation and consolidation in mammals remains largely unknown. Nevertheless, PKCδ is highly expressed in brain regions associated with learning, and more specifically aversive learning and memory (*Zafiri and Duvarci, 2022*). In particular, it is found in the CA3 layer of the hippocampus, as well as in a specific region of the central nucleus of the amygdala (CeA) called CE1 (*Haubensak et al., 2010*; see also https://mouse.brain-map.org/). There, PKCδ marks about 50% of CEl GABAergic neurons (*Haubensak et al., 2010*). Strikingly, it was shown that CeA is implicated in

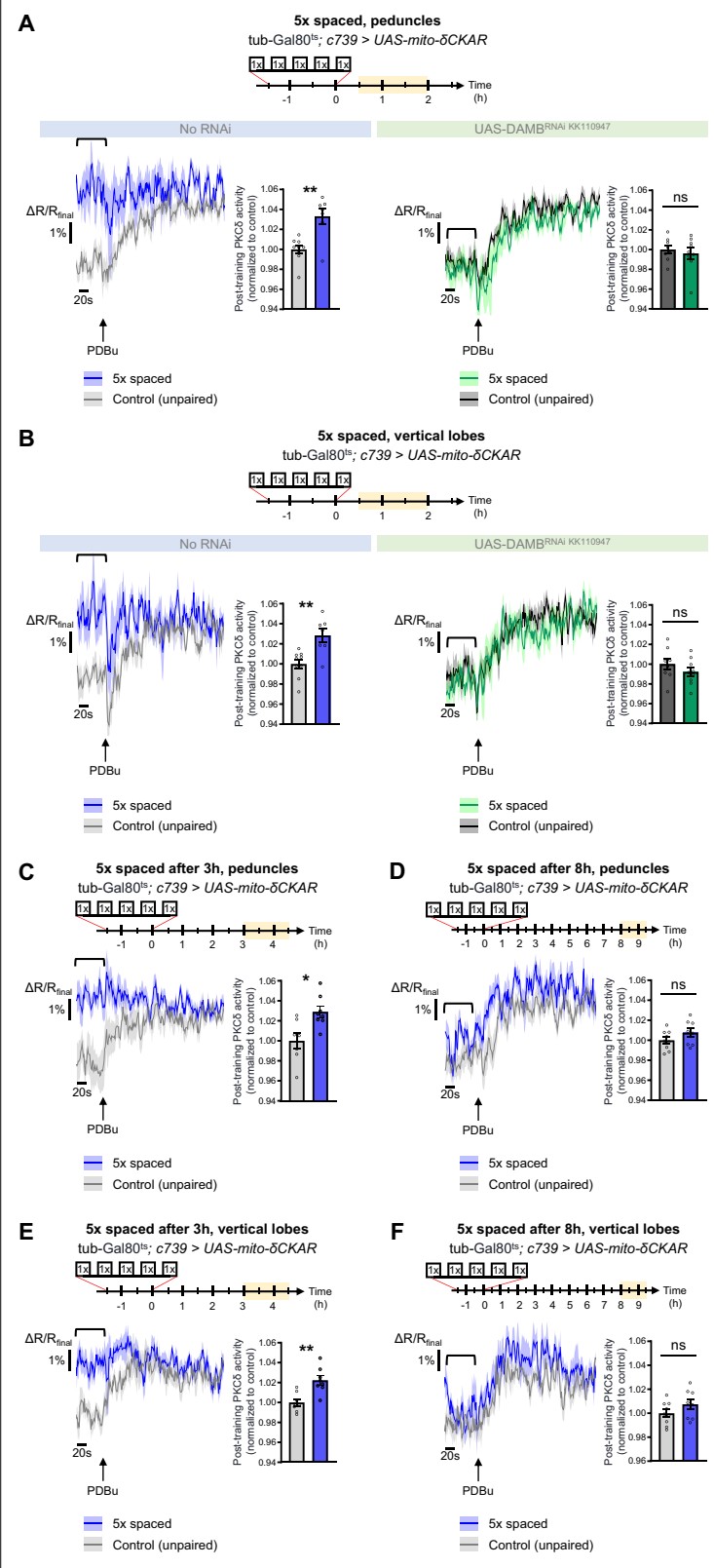

**Figure 5.** PKCδ translocates to the mitochondria of α/β neurons upon LTM formation. The mito-δCKAR sensor was expressed in adult α/β neurons and the post-training activity of mitochondrial PKCδ was measured in the peduncle (**A**) or vertical lobes (**B**) in the same flies, between 30 min and 2 hr post-conditioning (in yellow on the imaging time frame). The same approach was used as in *Figure 1F–H* with PDBu application to reach saturation

*Figure 5 continued*

level of the sensor and quantification of the mean post-training PKCδ activity performed on a time-window of 120 s before PDBu application (black line). (**A**) After 5 x spaced paired conditioning, mitochondrial PKCδ activity in the peduncle was increased as compared to unpaired conditioning (left panel, n=7–9, $t_{14}$=3.56, p=0.0032). When DAMB was knocked down in adult α/β neurons, mitochondrial PKCδ activity was not changed after 5 x spaced paired conditioning as compared to unpaired conditioning (right panel, n=8–9, $t_{15}$=1.09, p=0.29). (**B**) At the level of the vertical lobes, after 5 x spaced paired conditioning, mitochondrial PKCδ activity was increased as compared to unpaired conditioning (left panel n=7–9, $t_{14}$=3.93, p=0.0015). When DAMB was knocked down in adult α/β neurons, mitochondrial PKCδ activity was not changed after 5 x spaced paired conditioning, as compared to unpaired conditioning (right panel n=8–9, $t_{15}$=0.52, p=0.61). (**C**) The post-training activity of mitochondrial PKCδ was measured at the level of the peduncle, between 3 hr and 4 hr 30 min after 5 x spaced conditioning. At that timepoint, PKCδ mitochondrial activity was still increased as compared to 5 x spaced unpaired conditioning (n=7–8, $t_{14}$=3.01, p=0.010). (**D**) However, 8 h to 9 h 30 min after 5 x spaced conditioning, PKCδ mitochondrial activity in the peduncle was not significantly different from its 5 x spaced unpaired control (n=8–9, $t_{15}$=1.02, p=0.33). (**E**) Similarly, in the α lobe, between 3 hr and 4 hr 30 min after 5 x spaced conditioning, PKCδ mitochondrial activity was still increased as compared to 5 x spaced unpaired conditioning (n=8, $t_{14}$=3.99, p=0.0014), whereas (**F**) 8 hr to 9 hr 30 min after 5 x spaced conditioning, PKCδ mitochondrial activity in the α lobe was not significantly different from its 5 x spaced unpaired control (n=8–9, $t_{15}$=1.40, p=0.18). Data are expressed as mean ± SEM with dots as individual values, and were analyzed by unpaired two-sided t-test. Asterisks refer to the p-value of the unpaired t-test comparison using the following nomenclature: *p<0.05, **p<0.01, ns: not significant, p>0.05. See also *Figure 5—figure supplement 1*.

The online version of this article includes the following source data and figure supplement(s) for figure 5:

**Source data 1.** Source data displayed on *Figure 5*.

**Figure supplement 1.** PKCδ also translocates to the mitochondria of the β lobe upon LTM formation.

**Figure supplement 1—source data 1.** Source data displayed on *Figure 5—figure supplement 1*.

learning and consolidation of Pavlovian fear conditioning (*Steinberg et al., 2020*; *Wilensky et al., 2006*), and more specifically that PKCδ+neurons of the CeA projecting to the substantia innominata (SI) bidirectionally modulate negative reinforcement learning (*Cui et al., 2017*). However, no link between the function of PKCδ itself and memory consolidation or metabolism has been established yet, as the PKCδ-expressing feature of these neurons was only used as a means to genetically target them in the aforementioned studies. Our work thus sketches a possibly conserved role of neuronal PKCδ in the context of memory formation and consolidation in mammals, which nevertheless remains to be investigated. A possible role of PKCδ in plasticity mechanisms in mice was substantiated by the recent observation, in CA1 pyramidal neurons from mice organotypic slices, that stimulation-induced DAG production downstream of NMDA receptors (*Colgan et al., 2018*) can activate PKCδ in spines for local plasticity, and, at a longer timescale, its translocation to the nucleus for plasticity-induced transcription upon LTP formation (*Colgan et al., 2023*).

How is post-learning dopamine signaling coupled with PKCδ activation? DAMB is a dopamine receptor that has the ability to activate Gq with great efficiency and dopamine sensitivity (*Himmelreich et al., 2017*), which results in the production of DAG, a canonical activator of the majority of PKC enzymes. Once activated by DAG, the local hydrophobic patch generated by DAG fixation is thought to guide PKCs to various membranes (*Hurley and Grobler, 1997*), and notably to mitochondria in the case of PKCδ where it localizes to the intermembrane space (*Acin-Perez et al., 2010*; *Kim and Hammerling, 2020*). In addition, PKCδ is specific among PKCs in that it does not require activation loop phosphorylation for catalytic competence (*Duquesnes et al., 2011*). This direct link between DAG and PKCδ (and the absence of any requirement for PKCδ phosphorylation) may explain its physiological role as an efficient relay between extracellular cues sensed by receptors and intracellular metabolism at the level of the mitochondria. DAMB/Gq can also activate calcium signaling (*Cassar et al., 2015*; *Himmelreich et al., 2017*), the other canonical second-messenger pathway downstream of Gq. Since increases in mitochondrial calcium flux are another way to enhance PDH complex activity (*Denton et al., 1972*; *Glancy and Balaban, 2012*), one cannot exclude that mitochondrial calcium signaling may also play a role in the gating mechanism for LTM formation.

Gq/PKC signaling has also been shown to play a major role in memory in rodents. Mutation in the phosphosite of TrkB that binds phospholipase C (PLC) to produce DAG and activate PKC signaling impaired expression and maintenance of synaptic long-term potentiation in the hippocampus, and

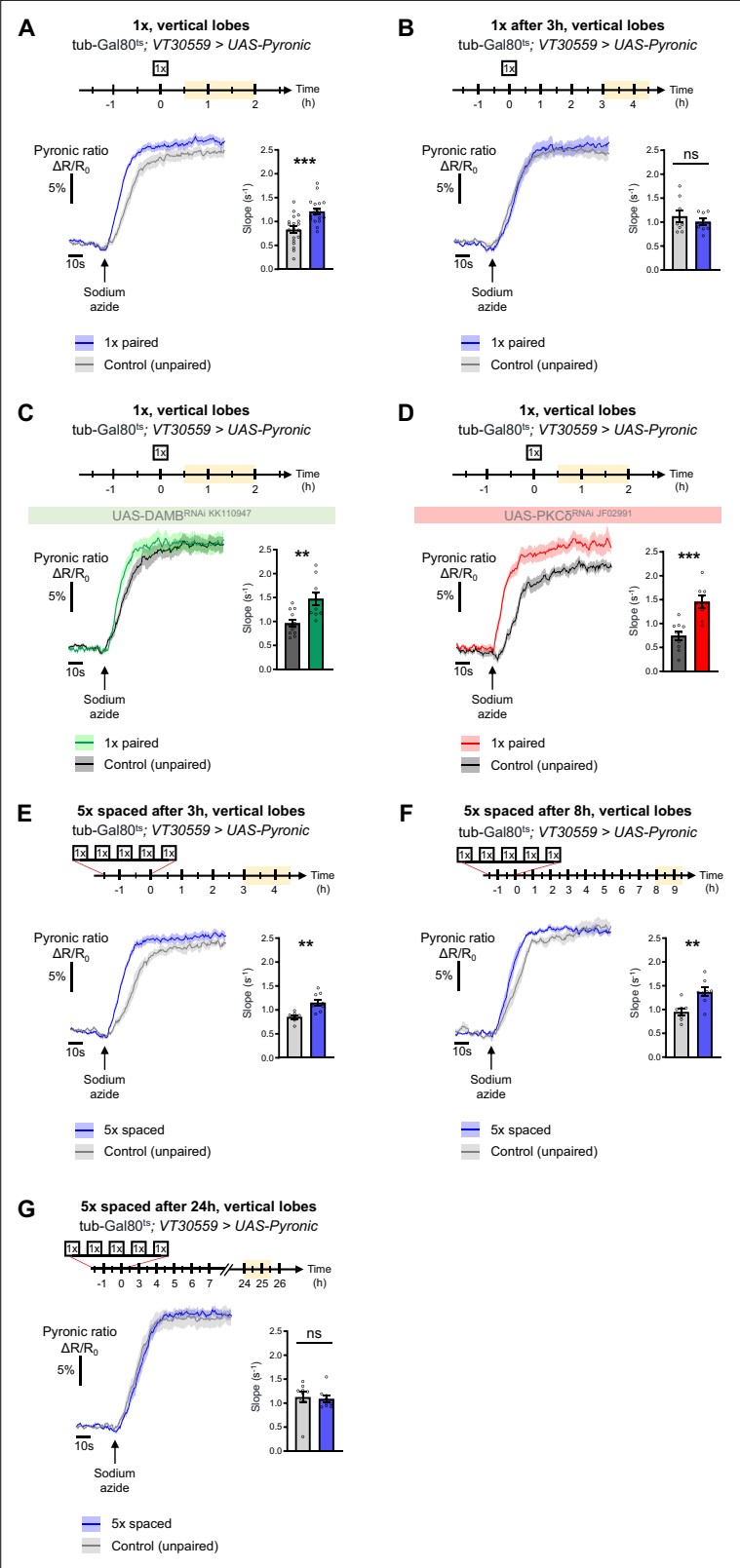

**Figure 6.** The DAMB/PKCδ axis is specific to metabolic activation during LTM formation. The pyruvate sensor Pyronic was expressed in adult MB neurons and pyruvate accumulation was measured after conditioning in the vertical lobes. For each panel, the period of imaging after conditioning is indicated in yellow on the imaging time frame. (**A**) 30 min to 2 hr after 1 x paired conditioning, an increased pyruvate flux was measured in control flies at

*Figure 6 continued on next page*

*Figure 6 continued*

the level of the vertical lobes, as compared to non-associative unpaired 1 x training (slope measurement n=17–20, $t_{35}$=4.08, p=0.0003). Here, we pooled the data obtained from control flies acquired in parallel to DAMB and PKCδ knockdown flies (panels C and D). (**B**) 3 hr to 4 hr 30 min after 1 x paired conditioning of control flies the rate of pyruvate accumulation was similar in the vertical lobes as compared to unpaired conditioning (slope measurement n=8, $t_{14}$=0.80, p=0.44). (**C**) 30 min to 2 hr after 1 x paired conditioning, the pyruvate flux was still increased when DAMB was knocked down in the MBs at the adult stage (slope measurement n=9–12, $t_{19}$=3.53, p=0.0022). (**D**) Similarly, PKCδ knockdown in adult MB neurons did not impair the 1 x conditioning induced increase in pyruvate accumulation in MB neuron axons (slope measurement n=8–10, $t_{16}$=4.57, p=0.0003). (**E**) 3 hr to 4 hr 30 min after the last cycle of 5 x spaced conditioning of control flies, the pyruvate flux was increased in the vertical lobes as compared to 5 x spaced unpaired conditioning (slope measurement n=9–7, $t_{14}$=3.89, p=0.0016). (**F**) 8 hr to 9 hr 30 min after the last cycle of 5 x spaced conditioning of control flies, the pyruvate flux was still increased in the vertical lobes as compared to 5 x spaced unpaired conditioning (slope measurement n=8, $t_{14}$=3.63, p=0.0027). (**G**) 24 hr after the last cycle of 5 x spaced conditioning of control flies, the pyruvate flux was similar as compared to 5 x spaced unpaired conditioning at the level of the vertical lobes (slope measurement n=9, $t_{16}$=0.33, p=0.75). Data are expressed as mean ± SEM with dots as individual values, and were analyzed by unpaired two-sided t-test. Asterisks refer to the p-value of the unpaired t-test comparison using the following nomenclature: **p<0.01, ***p<0.001, ns: not significant, p>0.05. See also *Figure 6—figure supplement 1*.

The online version of this article includes the following source data and figure supplement(s) for figure 6:

**Source data 1.** Source data displayed on *Figure 6*.

**Figure supplement 1.** Additional characterization of the temporal dynamic of the pyruvate flux following conditioning in the somas and medial lobes.

**Figure supplement 1—source data 1.** Source data displayed on *Figure 6—figure supplement 1*.

overall reduced learning in mice (*Gruart et al., 2007*; *Minichiello et al., 2002*). Thus, the Gq-DAG mechanism of PKCδ activation for memory consolidation could be conserved in mammals. While DAMB has no direct homology with mammalian dopaminergic receptors (instead belonging to an 'invertebrate type' class of receptors as demonstrated by sequence comparison *Mustard et al., 2005*), it should be noted that several mammalian dopamine receptors are also known to mobilize Gq signaling, such as the D1-D2 and the D5-D2 dopamine receptor heteromers (*So et al., 2009*; *Young and Thomas, 2014*). D1-like receptor can also directly engage with Gq/11 and therefore PLC (*Rashid et al., 2007*). In mammals, this non-canonical D1-like receptor/Gq coupling was notably found in the hippocampus and amygdala (*Jin et al., 2001*; *Ming et al., 2006*), two brain areas that are enriched in PKCδ+neurons (*Haubensak et al., 2010*; *Zafiri and Duvarci, 2022*). Anatomical studies demonstrate that both D1 and D2 receptors are expressed in the CeA, a region of the brain enriched in PKCδ+neurons (*Zafiri and Duvarci, 2022*). Periaqueductal gray/dorsal raphe (PAG/DR) afferent dopaminergic neurons exhibit phasic activation in response to an aversive unconditioned stimulus (US) and to a conditioned stimulus (CS) associated with the aversive US (*Groessl et al., 2018*; *Zafiri and Duvarci, 2022*), which could preferentially activate the D1 receptor-expressing CeA neurons during aversive learning (*Zafiri and Duvarci, 2022*). It would be of particular interest to explore the involvement of PKCδ neurons in this process, and to study whether PKCδ could be mobilized downstream of D1/Gq signaling in the context of aversive memory in mammals. Aside from dopaminergic receptors, a wide variety of memory-relevant receptors have been reported to activate Gq signaling, with adrenergic receptors being the most commonly cited kind. In physiological conditions, α-adrenergic receptors are considered to be the principal activators of Gq signaling (*Duquesnes et al., 2011*). β-adrenoreceptor subtypes are expressed in the hippocampus and amygdala; these regions receive noradrenergic afferences from the locus coeruleus, which plays a critical role in regulating behavioral memory in rodents, and are key for the processing of memories (*Sara, 2009*), and more specifically for their consolidation (*Sara, 2009*; *Souza-Braga et al., 2018*).

Here, we propose a model in which a state of high energy consumption in MB neuron axons necessary for LTM gating is prolonged by PKCδ, which boosts the PDH complex activity in MB mitochondria. Such a role is consistent with previous descriptions in non-neuronal mammalian cells. It is indeed known that PKCδ can regulate the PDH complex (*Acin-Perez et al., 2010*; *Kim and Hammerling, 2020*): activation of PKCδ leads to the phosphorylation of a putative PDK phosphatase, thereby enhancing its activity and promoting dephosphorylation of PDK, which inhibits it and thereby releases the inhibition it exerts on the PDH complex (*Acin-Perez et al., 2010*; *Kim and*

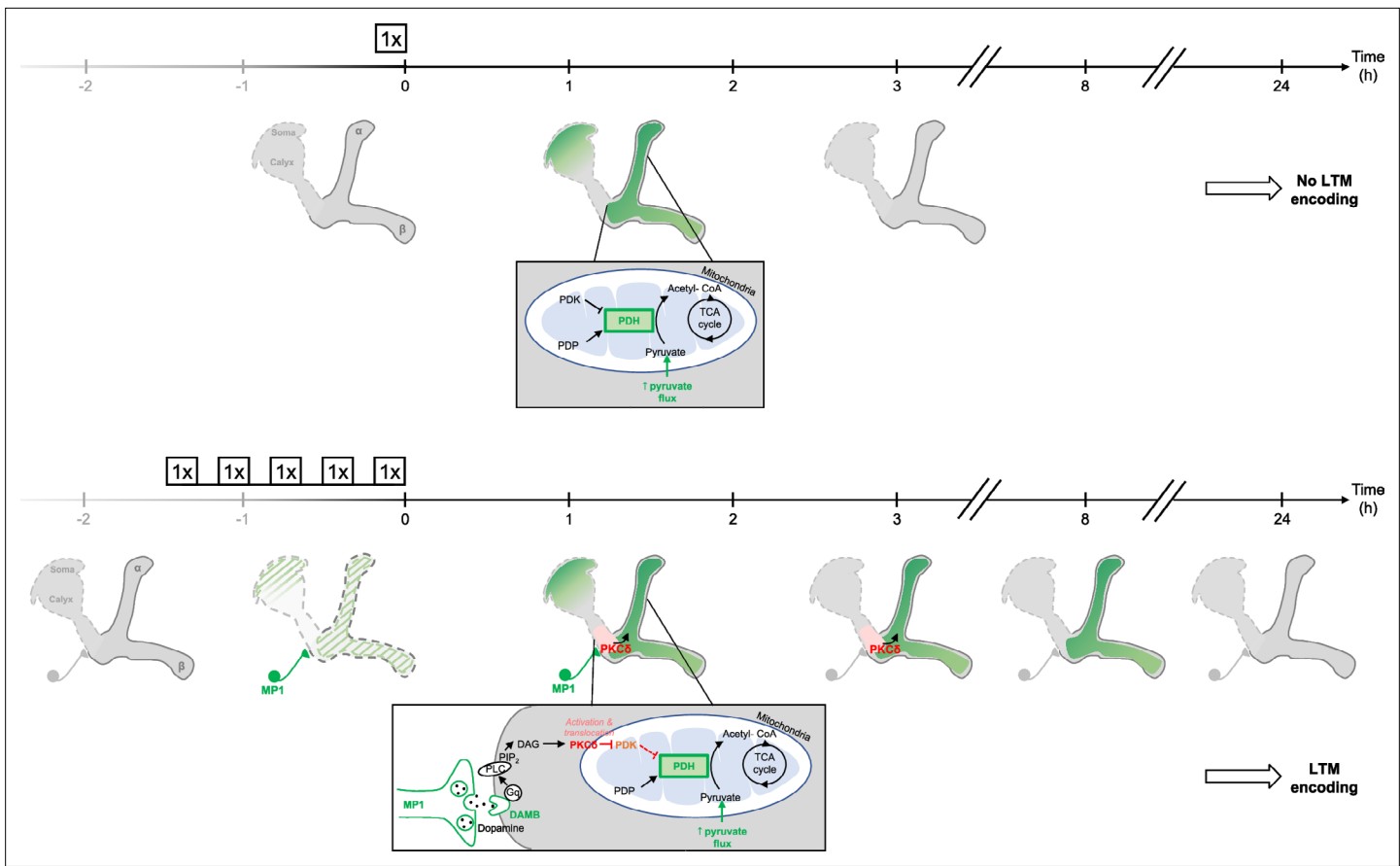

**Figure 7.** Schema of the energy-based gating of LTM, illustrating the role of PKCδ downstream of the DAMB signaling cascade. After 1 x training, the pyruvate flux to the mitochondria in the MB neurons somas, the vertical lobes (*Rabah et al., 2023*), and in the medial lobes is increased. This increased energy state is not maintained, and after 3 hr, the MBs are back to their basal energy state. Upon spaced training, in addition to the mechanism occurring after 1 x training, the enhanced oscillatory activity of MP1 neurons activates DAMB signaling in the efferent α/β MB neurons. The DAMB receptor preferentially couples with Gq and the produced DAG activates PKCδ, which results in its translocation to mitochondria. There, PKCδ can activate the PDH, by releasing the inhibition exerted by PDK. The enhanced PDH activity promotes an increase in mitochondrial pyruvate flux, and this enhanced energy state gates memory consolidation, thereby enabling LTM formation. After 3 hr, the lasting effect of activated PKCδ on mitochondrial metabolism maintains a high pyruvate flux in the vertical and medial lobes, while it comes back to its baseline levels in the somas. After 8 hr, whereas PKCδ activation has ended, the high energy state of the MBs is still maintained, which ultimately results in the gating of LTM. At 24 hr, once LTM has been encoded and can be tested by behavioral assays, the MB neurons mitochondrial metabolism is back to its basal state, indicating that the upregulation of pyruvate metabolism does not represents the memory trace in itself.

*Hammerling, 2020*). As PDK knockdown rescued the LTM defect induced by PKCδ knockdown in our study, we propose that PKCδ boosts the PDH complex activity via this mechanism of PDK-Pase activation, resulting in PDK inhibition in MB neuronal mitochondria. This lasting effect of PKCδ on mitochondrial metabolism thus maintains the pyruvate flux upregulated in the mitochondria of the MBs lobes, which unlocks memory consolidation to form LTM. The temporality of these successive events following spaced training leading to the gating of LTM is also particularly interesting: first, the MP1 neurons fire for up to 2 hr, which activates PKCδ for more than 3 hr (*Figure 5C and E*) – once phosphorylated, PKCδ remain active until dephosphorylation by phosphatases (*Kajimoto et al., 2010*). Mitochondrial metabolism is then upregulated for more than 8 hr (*Figure 6F*, *Figure 6—figure supplement 1F*), even though PKCδ activity level is back to its baseline level at that time-point (*Figure 5D and F*), which indicates that additional sustaining mechanisms relay PKCδ boosting effect on mitochondrial metabolism. For instance, one could speculate that mitochondrial motility, that is activated in the hours following spaced conditioning (*Pavlowsky et al., 2024*) alongside PKCδ activation and metabolic upregulation, could account for that long-term increase of the MB axons metabolic state by importing increasingly more mitochondria into the MB axons, while the

resulting decrease in the number of mitochondria in the somas would therefore render them unable to maintain metabolic activation beyond 3 hr in this compartment (*Figure 6—figure supplement 1B*).

While our work uncovers a molecular switch controlling the metabolic upregulation of the MB neurons, the role for this enhanced pyruvate flux to mitochondria has not yet been unveiled. Several hypotheses could be considered, which are not mutually exclusive. First, increased pyruvate incorporation into the TCA cycle could lead to acceleration of oxidative phosphorylation and subsequent enhanced ATP production, in support of sustained neuronal activity or of the substantial energy cost of de novo protein synthesis on which LTM formation depends at later stages (*Davis and Squire, 1984*; *Helmstetter et al., 2008*; *Jarome and Helmstetter, 2014*). Second, acceleration of the TCA cycle and oxidative phosphorylation can also be accompanied by increased ROS production (*Rosato et al., 2014*), both of which are increasingly recognized as neuronal signaling molecules (*Sinenko et al., 2021*; *Zhang et al., 2016*). Third, acceleration of the TCA cycle alone, decoupled from oxidative phosphorylation, could produce acetylcholine, the neurotransmitter used by MB neurons, and/or acyl groups that are crucial for subsequent epigenetic modifications. This non-canonical TCA cycle has recently been reported in a model of mammalian stem cells (*Arnold et al., 2022*). Finally, TCA cycle intermediates could be used for amino acid synthesis to fulfill the need for de novo protein synthesis, which is a hallmark of LTM.

Our study extends to the context of memory formation in neurons the beneficial role of PKCδ in the modulation of mitochondrial metabolism, whereas it was first extensively described in the mouse embryonic fibroblast (*Acin-Perez et al., 2010*). There, PKCδ serves as an indispensable supervisor of mitochondrial energy production: by sensing the cytochrome c redox state as a proxy of the respiratory chain workload, it adjusts mitochondrial metabolism within safe margins (that can be overpassed when fuel flux surpasses the capacity of the electron-transport chain, causing detrimental ROS release; *Acin-Perez et al., 2010*; *Kim and Hammerling, 2020*). Strikingly, PKCδ was also widely described as detrimental to cells in pathological conditions, with a well-characterized role in cardiac (*Inagaki et al., 2003*; *Zaja et al., 2014*) and neuronal (*Bright et al., 2004*; *Dave et al., 2011*; *Phan et al., 2002*) cell death upon ischemic incidents. In this model, PKCδ takes part in apoptotic events through a caspase-induced signaling cascade in mitochondria, notably via cleavage by caspase-3 at the PKCδ caspase-dependent cleavage site, which releases a 40 kDa active fragment. Furthermore, transient episodes of ischemia can also activate PKCδ through ROS-dependent mechanisms, leading to its phosphorylation and subsequent activation. PKCδ can then bind and phosphorylate Drp1, a major mitochondrial fission protein, thereby increasing its activity. This mechanism was described in cardiomyocytes during anoxia-reoxygenation injury (*Zaja et al., 2014*), and in neurons under oxidative stress in cellulo, as well as in vivo in the context of hypertension-induced encephalopathy (*Qi et al., 2011*). Finally, PKCδ has been implicated in neurodegenerative diseases such as Alzheimer's disease (AD) (*Du et al., 2018*) and Parkinson's disease (PD) (*Kaul et al., 2005*; *Yang et al., 2004*; *Zhang et al., 2007*). However, a number of these studies should be considered carefully, as some of their results are based on PKCδ inhibition by rottlerin, which has been found to be an inappropriate and ineffective PKCδ pharmacological inhibitor (*Soltoff, 2007*). Nevertheless, various pieces of evidence show increased expression of PKCδ in AD patients (*Du et al., 2018*), pointing towards the fact that PKCδ inhibition plays an important protective role in brain aging (*Conboy et al., 2009*), ischemia (*Bright et al., 2004*; *Dave et al., 2011*; *Phan et al., 2002*), and neurodegenerative disease (*Du et al., 2018*; *Kaul et al., 2005*; *Yang et al., 2004*; *Zhang et al., 2007*).

Altogether, it is now emerging that PKCδ is functionally ambivalent: while its role in metabolism regulation maintains essential physiological functions such as LTM gating in the brain as demonstrated in this study, stress conditions unveil negative effects mediated by this kinase that derail the system equilibrium. The interplay between the two modes of activation of PKCδ by DAG and oxidative stress is probably at the heart of its contrasting duality. Expanding upon our work, it could be of particular interest to explore whether competition between PKCδ activation via these two modes, DAG and oxidative stress, occurs upon aging and/or pathology.

# Materials and methods

## Key resources table

| Reagent type (species) or resource | Designation | Source or reference | Identifiers | Additional information |
|---|---|---|---|---|
| Genetic reagent (*D. melanogaster*) | tubulin-Gal80ts;VT30559-Gal4 | *Plaçais et al., 2017* | N/A | Available upon request |
| Genetic reagent (*D. melanogaster*) | tubulin-Gal80ts;c739-Gal4 | *Turrel et al., 2018* | N/A | Available upon request |
| Genetic reagent (*D. melanogaster*) | tubulin-Gal80ts;elav-Gal4 | *Silva et al., 2022* | N/A | Available upon request |
| Genetic reagent (*D. melanogaster*) | 30E11-LexA | Bloomington *Drosophila* Stock Center | BDSC:54209; FLYB: FBst0054209; RRID:BDSC_54209 | FlyBase symbol: P{GMR30E11-lexA} |
| Genetic reagent (*D. melanogaster*) | UAS-PKCδ$^{RNAi\ JF02991}$ | Bloomington *Drosophila* Stock Center | BDSC:28355; FLYB: FBst0028355; RRID:BDSC_28355 | FlyBase symbol: P{TRiP.JF02991} |
| Genetic reagent (*D. melanogaster*) | UAS-PKCδ$^{RNAi\ KK109117}$ | Vienna *Drosophila* Resource Center | VDRC:101421; FLYB: FBti0121612; RRID:VDRC_101421 | FlyBase symbol: P{KK109117} |
| Genetic reagent (*D. melanogaster*) | UAS-DAMB$^{RNAi\ KK110947}$ | Vienna *Drosophila* Resource Center | VDRC:105324; FLYB: FBst0477151; RRID:VDRC_105324 | FlyBase symbol: P{KK110947} |
| Genetic reagent (*D. melanogaster*) | UAS-PDK$^{RNAi\ KK106641}$ | Vienna *Drosophila* Resource Center | VDRC:106641; FLYB: FBst0478465; RRID:VDRC_106641 | FlyBase symbol: P{KK107950} |
| Genetic reagent (*D. melanogaster*) | LexAop-dTrpA1 | *Liu et al., 2012* | N/A | Available upon request |
| Genetic reagent (*D. melanogaster*) | tubulin-Gal80ts;VT30559-Gal4, UAS-Pyronic | *Plaçais et al., 2017* | N/A | Available upon request |
| Genetic reagent (*D. melanogaster*) | 30E11-LexA;VT30559-Gal4 | *Plaçais et al., 2017* | N/A | Available upon request |
| Genetic reagent (*D. melanogaster*) | UAS-cyto-δCKAR | This paper | N/A | Available upon request; see "generation of transgenic flies" in Material and methods |
| Genetic reagent (*D. melanogaster*) | UAS-mito-δCKAR | This paper | N/A | Available upon request; see "generation of transgenic flies" in Material and methods |
| Genetic reagent (*D. melanogaster*) | UAS-cyto-δCKAR;UAS-PKCδ$^{RNAi\ JF01991}$ | This paper | N/A | Available upon request |
| Genetic reagent (*D. melanogaster*) | UAS-mito-δCKAR;UAS-DAMB$^{RNAi\ KK110947}$ | This paper | N/A | Available upon request |
| Genetic reagent (*D. melanogaster*) | UAS-mito-δCKAR;UAS-DAMB$^{RNAi\ KK110947}$;LexAop-dTrpA1 | This paper | N/A | Available upon request |
| Genetic reagent (*D. melanogaster*) | UAS-PDK$^{RNAi\ KK106641}$;UAS-PKCδ$^{RNAi\ JF02991}$ | This paper | N/A | Available upon request |
| Genetic reagent (*D. melanogaster*) | UAS-mito-DsRed | Bloomington *Drosophila* Stock Center | BDSC:93056; FLYB: FBst0093056; RRID:BDSC_93056 | FlyBase symbol: P{UAS-DsRed.mito} |
| Genetic reagent (*D. melanogaster*) | UAS-mito-δCKAR;UAS-mito-DsRed | This paper | N/A | Available upon request |
| Recombinant DNA reagent | pcDNA3-deltaCKAR | Addgene | #31526 RRID:Addgene_31526 | |
| Recombinant DNA reagent | pJFRC-MUH | Addgene | #26213 RRID:Addgene_26213 | |
| Recombinant DNA reagent | pJRFC-MUH-UAS-deltaCKAR | This paper | N/A | cyto-δCKAR construct used to generate the UAS-cyto-δCKAR *Drosophila* line |
| Recombinant DNA reagent | pJRFC-MUH-UAS-mitodeltaCKAR | This paper | N/A | mito-δCKAR construct used to generate the UAS-mito-δCKAR *Drosophila* line |
| Sequence-based reagent | Primer PKCδ forward | This paper, DRSC Fly Primer Bank | PCR Primer Pair PP14953 | 5'-GGCACCAAACACCCGTATCT-3' |
| Sequence-based reagent | Primer PKCδ reverse | This paper, DRSC Fly Primer Bank | PCR Primer Pair PP14953 | 5'-CCCATAGAATCTGGCTCGCT-3' |
| Sequence-based reagent | Primer PDK forward | This paper, DRSC Fly Primer Bank | PCR Primer Pair PP14510 | 5'-CCTCGCCCCTCTCGATAAAG-3' |
| Sequence-based reagent | Primer PDK reverse | This paper, DRSC Fly Primer Bank | PCR Primer Pair PP14510 | 5'-TCGAACAGGCAGTTCCTTGC-3' |

*Continued on next page*

*Continued*

| Reagent type (species) or resource | Designation | Source or reference | Identifiers | Additional information |
|---|---|---|---|---|
| Sequence-based reagent | Primer tub forward | *Turrel et al., 2018* | N/A | 5'-TTGTCGCGTGTGAAACACTTC-3' |
| Sequence-based reagent | Primer tub reverse | *Turrel et al., 2018* | N/A | 5'-CTGGACACCAGCCTGACCAAC-3' |
| Commercial assay or kit | RNeasy Plant Mini Kit | QIAGEN | Cat. #74904 | |
| Commercial assay or kit | RNA MinElute Cleanup Kit | QIAGEN | Cat. #74204 | |
| Commercial assay or kit | SuperScript III First-Strand Kit | Thermofisher Invitrogen | Cat. #18080–051 | |
| Commercial assay or kit | SYBR Green I Master mix | Roche | Cat. # 04729692001 | |
| Chemical compound, drug | 3-octanol (99%) | Sigma-Aldrich | Cat. #153095 | |
| Chemical compound, drug | 4-methylcyclohexanol (98%) | Sigma-Aldrich | Cat. #218405 | |
| Chemical compound, drug | Paraffine GPR Rectapur | VWR | Cat. #24679.360 | |
| Chemical compound, drug | NaCl | Sigma-Aldrich | Cat. #S9625 | |
| Chemical compound, drug | KCl | Sigma-Aldrich | Cat. #P3911 | |
| Chemical compound, drug | $MgCl_2$ | Sigma-Aldrich | Cat. #M9272 | |
| Chemical compound, drug | $CaCl_2$ | Sigma-Aldrich | Cat. #C3881 | |
| Chemical compound, drug | D-trehalose | Sigma-Aldrich | Cat. #9531 | |
| Chemical compound, drug | Sucrose | Sigma-Aldrich | Cat. #S9378 | |
| Chemical compound, drug | HEPES-NaOH | Sigma-Aldrich | Cat. #H7637 | |
| Chemical compound, drug | Phorbol 12,13-dibutyrate (PDBu) | Tocris | Cat. #4153 | |
| Chemical compound, drug | Bisindolylmaleimide IV (Bis IV) | Sigma-Aldrich | Cat. #B3306 | |
| Chemical compound, drug | Sodium Azide | Sigma-Aldrich | Cat. #71289 | |
| Chemical compound, drug | Phosphate Buffered Saline (PBS) | Sigma-Aldrich | Cat. #P4417 | |
| Chemical compound, drug | Paraformaldehyde 16% | Life technologies | Cat. #P36965 | |
| Software, algorithm | Prism 8 | GraphPad software, v8.4.3 | RRID:SCR_002798 | |
| Software, algorithm | Fiji | ImageJ 1.52 p | RRID:SCR_002285 | |
| Software, algorithm | Affinity Photo | Affinity Photo software, v1.10.5 | RRID:SCR_016951 | |

## Resource availability

### Materials availability

Materials generated in this study are available from the corresponding authors without restriction.

## Experimental model and subject details

*D. melanogaster* flies were maintained on standard cornmeal-yeast-agar medium. Stocks were kept at 18 °C and 60% humidity under a 12 hr light:12 hr dark cycle. Genetic crosses were performed at 18 °C for behavior experiments or at 23 °C for imaging experiments (except when indicated otherwise). Both male and female flies were used for behavior experiments. Female flies were used for imaging experiments due to their larger size. Flies from the Vienna *Drosophila* Resource Center (VDRC) collection were outcrossed for five generations to a reference strain carrying the w[1118] mutation in an otherwise Canton Special (Canton S) genetic background. Since TRiP RNAi transgenes do

not carry a mini-white marker but are labeled with a y[+] marker, flies from the TRiP RNAi collection were outcrossed to a y[1]w[67c23] strain in an otherwise Canton S background. The Canton S strain was used as the wild type strain.

To target MB neurons, the VT30559-Gal4 line was used in combination with the thermosensitive TARGET system tubulin-Gal80[ts] to generate the tub-Gal80[ts];VT30559-Gal4 inducible driver line (*Plaçais et al., 2017*), so as to temporally control the expression of the desired transgene. Gal4 activity was released by transferring 0- to 2-day-old adult flies to 30 °C for 2–3 days. Likewise, the c739-Gal4 line was used to specifically target α/β neurons, and the tubulin-Gal80[ts];c739-Gal4 construct line previously generated in the laboratory and described in *Turrel et al., 2018* allowed temporal control of the Gal4 activity. As described in *Silva et al., 2022*, the tubulin-Gal80[ts];elav-Gal4 system was used for time-controlled pan-neuronal knockdown. To target MP1 neurons independently of the Gal4 system, the 30E11-LexA driver was used (*Plaçais et al., 2017*). The tub-Gal80[ts]; VT30559-Gal4, UAS-Pyronic line was previously generated in our research group and described in *Plaçais et al., 2017*. The UAS-cyto-δCKAR and UAS-mito-δCKAR lines were generated for this study (see 'Generation of transgenic flies' section below). The following genetic constructs were generated in this study by combining the appropriate UAS-RNAi line (listed in the key resources table) and/or FRET sensor transgenes: (i) UAS-cyto-δCKAR;UAS-PKCδ[RNAi JF01991], (ii) UAS-mito-δCKAR;UAS-DAMB[RNAi KK110947], (iii) UAS-mito-δCKAR;UAS-DAMB[RNAi KK110947];LexAop-dTrpA1, (iv) UAS-PDK[RNAi KK106641];UAS-PKCδ[RNAi JF02991], (v) UAS-mito-δCKAR;UAS-mito-DsRed. All other fly lines used in this study are listed in the key resources table and either come from the VDRC collection or TRiP RNAi collection, or were previously published.

## Method details

### Aversive olfactory conditioning and memory test

The behavioral experiments, including sample sizes, were conducted similarly to previous studies from our research group (*Plaçais et al., 2017*; *Scheunemann et al., 2018*). For all experiments, training and testing were performed in a sound- and odor-proof room at 25 °C and 80% humidity. Experimental flies (male and female) were transferred to fresh bottles containing standard medium on the day before conditioning for the non-induced condition. For the induced condition, flies were transferred 2 days before the experiment at 30.5 °C to allow RNAi expression.

### Conditioning

Flies were conditioned by exposure to one odor paired with electric shocks and subsequent exposure to a second odor in the absence of shock. A barrel-type machine was used for simultaneous automated conditioning of six groups of 40–50 flies each. Throughout the conditioning protocol, each barrel was attached to a constant air flow at 2 L.min[−1]. The odorants 3-octanol (odor O) and 4-methylcyclohexanol (odor M), diluted in paraffin oil at 0.360 and 0.325 mM respectively, were alternately used as conditioned stimuli (CS). For a single cycle of associative training, flies were first exposed to an odorant (the CS+) for 1 min while 12 pulses of 5 s-long, 60 V electric shocks were delivered; flies were then exposed 45 s later to a second odorant without shocks (the CS–) for 1 min. Here, the groups of flies were subjected to one of the following olfactory conditioning protocols: 1 cycle training (1 x), five consecutive associative training cycles (5 x massed training), or five associative cycles spaced by 15 min inter-trial intervals (5 x spaced conditioning). Non-associative control protocols (unpaired protocols) were also employed for in vivo imaging experiments. During unpaired conditionings, the odor and shock stimuli were delivered separately in time, with shocks occurring 3 min before the first odorant. After training and until memory testing, flies were kept on regular food at 25 °C (for 3 hr-memory test) or at 18 °C (for 24 hr-memory test).

### Memory test

The memory test was performed either 3 hr after 1 x conditioning or 24 hr after 5 x conditioning in a T-maze apparatus comprising a central elevator to transfer the flies to the center of the maze arms. During the test, flies were exposed simultaneously to both odors (the same concentration as during conditioning) in the T-maze. After 1 min of odorant exposure in the dark, flies were trapped in either T-maze arm, retrieved and counted. A memory score was calculated as the number of flies avoiding the conditioned odor minus the number of flies preferring the conditioned odor, divided by the total number of flies. A single memory score value is the average of two scores obtained from two groups of

genotypically identical flies conditioned in two reciprocal experiments, using either odorant (3-octanol or 4-methylcyclohexanol) as the CS+. The indicated 'n' is the number of independent memory score values for each genotype.

### Odor perception test

The olfactory acuity of flies was tested after conditioning with the CS+, since electric shocks modify their olfactory perceptions. Flies were then immediately tested in a T-maze, where they had to choose between the CS– or its solvent (paraffin oil). Odor concentrations used in this assay were the same as for the memory assays. At these concentrations, both odorants are innately repulsive. The odor-interlaced side was alternated for successively tested groups. After 1 min, flies were counted, and naive odor avoidance was calculated as for the memory test.

### Electric shock perception test

During the test, flies must choose between two barrels: one delivering the electric shocks, and one that is neutral. The compartment where the electric shocks are delivered was alternated between two consecutive groups. After 1 mine, flies were counted, and shock avoidance was calculated as for the memory test.

### In vivo imaging

Crosses for imaging experiments were raised at 23 °C and fly progeny were induced for 3 days at 30.5 °C to drive sufficient expression of the probe (and the desired RNAi) for use in imaging, except for MP1 activation experiments involving the thermosensitive LexAop-dTrpA1 transgene in which flies were always kept at 18 °C. As in all previous imaging work from our laboratory, all in vivo imaging was performed on female flies, which are preferred since their larger size facilitates surgery. Naive (not trained) or conditioned flies (1 x, 5 x spaced or 5 x massed and their corresponding unpaired controls) were gently handled by aspiration without anesthesia and glued on their dorsal side to a plastic coverslip coated with a thin transparent plastic sheet. The coverslip was then placed on a recording chamber. Surgery was performed to obtain an imaging window on the fly head by removing the cuticle, trachea and fat bodies, thereby exposing the underlying MB neurons. During the procedure, the head capsule is bathed in a drop of artificial hemolymph: NaCl 130 mM (Sigma cat. # S9625), KCl 5 mM (Sigma cat. # P3911), MgCl2 2 mM (Sigma cat. # M9272), CaCl2 2 mM (Sigma cat. # C3881), D-trehalose 5 mM (Sigma cat. # 9531), sucrose 30 mM (Sigma cat. # S9378), and HEPES hemisodium salt 5 mM (Sigma cat. # H7637). At the end of the procedure, any remaining solution was absorbed and a fresh 90 µL droplet was applied on the preparation. When performed on flies that underwent olfactory conditioning, the imaging experiments were performed within the time period after conditioning indicated on each figure panel.

### FRET δCKAR imaging

Imaging was performed using a SP8 DIVE Leica 2-photon microscope equipped with a 25 x, 1.0 NA water immersion objective. Two-photon excitation of CFP was achieved using an Insight X3 Spectra Physics laser tuned to 840 nm. 512x250 images were acquired at the rate of one image every 2 s, with two z plans imaged in parallel (vertical lobes and peduncle). Typically, the images comprised the structures of both brain hemispheres, although only one hemisphere was visible in some preparations. Two spectrally tunable hybrid detectors were adjusted to detect 440–490 nm CFP emission and 510–550 nm YFP emission. Two minutes after the beginning of image acquisition, 10 µL of a 2.5 mM phorbol 12,13-dibutyrate (PDBu, Tocris cat. # 4153) solution (25 mM PDBu stock solution in DMSO dissolved at 1/10 in artificial hemolymph) were injected into the 90 µL-droplet bathing the fly's brain, bringing PDBu to a final concentration of 250 µM (DMSO: 1/100). In control experiments, PDBu injection was replaced by the injection of DMSO alone (final concentration: 1/100). To inhibit PKCδ, two minutes after the beginning of image acquisition, 10 µL of a 50 µM bisindolylmaleimide IV (Bis IV, Sigma cat. # B3306) solution (500 µM stock solution in DMSO dissolved at 1/10 in artificial hemolymph), bringing Bis IV to a final concentration of 5 µM. In control experiments, Bis IV injection was replaced by the injection of DMSO alone (final concentration: 1/100). Drug application during the recording could give rise to artifactual perturbation of the signal at the time of injection, with variability from one experiment to another. In *Figure 1F and a* section of the presented traces

corresponding to the 15 s following the injection was smoothed using a running average procedure (with a time window of 30 s) to remove injection artefacts that were especially large in this initial series of experiments. The injection technique was subsequently improved so that no smoothing was applied on any of the other presented experiments (including in particular *Figure 2B*, which reports a similar phenomenon as *Figure 1F*).

For image analysis, regions of interest (ROI) were delimited by hand around each visible vertical lobe or peduncle region, and the average intensity of both CFP and YFP channels over each ROI were calculated over time after background subtraction; the background was evaluated as the mean intensity over a region of interest placed in a nonfluorescent part of the brain. The δCKAR sensor was designed so that FRET from CFP to YFP decreases when PKCδ phosphorylating activity increases. The inverse FRET ratio, ΔR (CFP/YFP), was calculated to obtain a signal that positively correlates with PKCδ phosphorylating activity. To measure the δCKAR response (*Figure 1D–E*, *Figure 1—figure supplement 1C–D*, *Figure 4*), the ΔR ratio was normalized by a baseline value calculated over the 2 min preceding drug injection (PDBu or Bis IV). To measure PKCδ post-training activity (*Figure 1F–H*, *Figure 2B–D*, *Figure 5*), the ΔR ratio was normalized by a plateau value calculated from 6 min 40 s to the end of the recording, and the controls were normalized to 1.

For the MP1 activation experiments in which PKCδ response activity was assayed (*Figure 4*), flies were prepared as previously reported and placed on a custom-made device equipped with a Peltier cell to allow precise control of the temperature under the microscope. Flies were recorded for 2 min at 18 °C to establish a baseline and then received thermal treatment for MP1 activation, consisting of three consecutive periods of 2 min at 30 °C followed by 2 min at 18 °C. The recording was continued for 10 min after the last activation period. To measure PKCδ response activity, the ΔR was normalized by the baseline value calculated over the 2 min preceding thermal treatment.

The indicated 'n' is the number of animals that were assayed in each condition.

## FRET imaging of pyruvate flux

Two-photon imaging was performed using a Leica TCS-SP5 upright microscope equipped with a 25 x, 0.95 NA water immersion objective. Two-photon excitation was achieved using a Mai Tai DeepSee laser tuned to 825 nm. The frame rate was two images per second. 512x150 images were acquired at a rate of two images per second. The emission channels for mTFP and Venus were the same as described in *Gervasi et al., 2010*. Measurements of pyruvate consumption were performed according to a previously well-characterized protocol (*Plaçais et al., 2017*). After 1 min of baseline acquisition, 10 μL of a 50 mM sodium azide solution (Sigma cat. #71289; prepared in the same artificial hemolymph solution) were injected into the 90 μL-droplet bathing the fly's brain, bringing sodium azide to a final concentration of 5 mM. In experiments involving pretreatment of the brains (*Figure 3A*), 250 μM of PDBu or 1/100 DMSO (final concentrations) were injected into the hemolymph droplet 3 mins prior to sodium azide injection. Image analysis was performed as previously described (*Plaçais et al., 2017*). ROI were delimited by hand around each visible MB vertical lobe, and the average intensity of the mTFP and Venus channels over each ROI was calculated over time after background subtraction. The Pyronic sensor was designed so that FRET from mTFP to Venus decreases when the pyruvate concentration increases. To obtain a signal that positively correlates with pyruvate concentration, the inverse FRET ratio was calculated as mTFP intensity divided by Venus intensity. This ratio was normalized by a baseline value calculated over the 1 min preceding drug injection. The slope was calculated between 10 and 70% of the plateau. The indicated 'n' is the number of animals that were assayed in each condition.

## Confocal imaging

Female flies carrying the VT30559-Gal4 MB neuron driver were crossed with males carrying the two transgenes UAS-mito-δCKAR;UAS-mito-DsRed. The cross was raised at 25 °C and the adult progeny was fixed overnight in 4% paraformaldehyde (Electron Microscopy Sciences, 15710) at 4 °C. Brains were dissected on ice in 1×PBS (Sigma cat. #P4417) and directly mounted using Prolong Mounting Medium (Life Technologies cat. #P36965). Next, z-stacks of the MB neurons' somas composed of 1,024×1,024 px images were acquired with a Nikon A1R confocal microscope equipped with a×100/1.40 oil-immersion objective, with a step of 1 μm between each plan. Confocal excitation of the mito-δCKAR YFP fluorophore was achieved using a laser tuned to 488 nm while DsRed was

excited using a 561 nm laser. The two detectors were adjusted to detect 515–530 nm YFP emission and 570–620 nm DsRed emission. Maximum intensity projection of the 28 images composing the z-stack covering the soma region was generated using Fiji (ImageJ 1.52 p), and the merged image of the two acquired channels was made using Affinity Photo v1.10.5.

## RT-qPCR analyses

To assess the efficiency of the PKCδ RNAi used in this study, female flies carrying the tubulin-Gal80[ts];elav-Gal4 pan-neuronal inducible driver were either crossed with UAS-PKCδ[RNAi JF02991] or UAS-PKCδ[RNAi KK109117] males, or with CS males for controls (please note that crosses of these RNAi lines with the constitutive elav-Gal4 driver were lethal for the progeny). To assess PKCδ presence in the MB neurons, female flies carrying the tubulin-Gal80[ts];VT30559-Gal4 MB neuron driver were either crossed with UAS-PKCδ[RNAi JF02991] males, or with CS males for controls. Fly progeny was raised at 23 °C throughout development. Newly hatched flies were transferred to fresh food vials at 30.5 °C for 4 days of induction before RNA extraction, as previously reported (*Silva et al., 2022*). To assess the efficiency of the PDK RNAi, female flies carrying the elav-Gal4 driver were either crosses with UAS-PDK[RNAi KK106641] males or with CS males for controls (the progeny was viable using this constitutive driver). Fly progeny was raised at 25 °C. RNA extraction and cDNA synthesis were performed using the RNeasy Plant Mini Kit (QIAGEN), RNA MinElute Cleanup Kit (QIAGEN), oligo(dT)20 primers and the SuperScript III First-Strand kit (Thermo Fisher Invitrogen). Amplification was performed using a LightCycler 480 (Roche) and the SYBR Green I Master mix (Roche). Specific primers used for PKCδ or PDK cDNA and the refence α-Tub84B (Tub, CG1913) cDNA are specified in the key resources table. The level of PKCδ cDNA was compared to the level of the α-Tub84B reference cDNA. Each reaction was performed in triplicate. The specificity and size of amplification products were assessed by melting curve analyses. Expression relative to the reference was presented as the foldchange compared to the average of control genotype groups measured in parallel (relative quantification $RQ = 2^{-\Delta\Delta Ct}$, where Ct is the cycle threshold). The entire data series were normalized to the control genotype.

## Generation of transgenic flies

To generate the UAS-cyto-δCKAR line, the pcDNA3-deltaCKAR plasmid (Addgene #31526) was digested by HindIII and XbaI. The resulting fragment was purified by electrophoresis and cloned into a pJFRC-MUH plasmid (Addgene #26213) in the XbaI/NotI subcloning site. The resulting construct was verified by restriction analysis and sequenced by PCR. This subcloning was outsourced to RDBio-tech (France). The UAS-mito-δCKAR line was generated using the plasmid mito-δCKAR generated in *Wu-Zhang et al., 2012* by subcloning the mito-δCKAR fragment (HindIII/XbaI) into the pJFRC-MUH plasmid. The generation of the transgenic fly strains via the embryonic injection of the two vectors was outsourced to Rainbow Transgenic Flies, Inc (CA, USA).

## Quantification and statistical analysis

Statistical parameters including the definitions and exact value of n, deviations and p values are reported in the figures and corresponding legends. Data are expressed as the mean ± SEM with dots as individual values corresponding to either a group of 40–50 flies analyzed together in a behavioral assay, or the response of a single recorded fly for imaging. Statistical analysis and graphs were made using Prism 8 (GraphPad software, v8.4.3). Comparisons between two groups were performed using a two-tailed unpaired t-test; results are provided as the value $t_x$ of the t distribution with x degrees of freedom obtained from the data. Comparisons between multiple groups were performed using one-way ANOVA followed by Tukey's post hoc test for pairwise comparisons. ANOVA results are given as the value of the Fisher distribution $F_{x,y}$ obtained from the data, where x is the numerator degrees of freedom and y is the denominator degrees of freedom. Asterisks denote the smallest significant

difference between the relevant group and its controls with the post hoc comparisons (*p<0.05, **p<0.01, ***p<0.001, ****p<0.0001, ns: not significant).

## Acknowledgements

We thank the TRiP consortium at Harvard Medical School and the BDSC and VDRC fly stock centers for providing transgenic RNAi fly stocks, and Eloïse de Trédern for help in subcloning the mito-δCKAR plasmid. We thank Alexandre Didelet and Christelle Beauchamp for technical support with fly food preparation. We are also grateful to Dr Jaime de Juan-Sanz for insightful comments on the manuscript. The authors acknowledge funding from the European Research Council (ERC Advanced Grant EnergyMemo n°741550, to TP), from the Agence Nationale de la Recherche (ANR n°20-CE92-0047-01, to P-YP) and from DIM ELICIT's grant from Région Ile-de-France (equipment grant, to P-YP). TC was funded by doctoral fellowships from the French Ministry of Research for three years, and from the Fondation pour la Recherche Medicale (FRM) for one year (grant number FDT202304016704).

## Additional information

### Funding

| Funder | Grant reference number | Author |
| --- | --- | --- |
| European Research Council | AdG-741550 | Thomas Preat |
| Agence Nationale de la Recherche | 20-CE92-0047-01 | Pierre-Yves Plaçais |
| Fondation pour la Recherche Médicale | FDT202304016704 | Typhaine Comyn |
| Region Ile-de-France | DIM ELICIT | Pierre-Yves Plaçais |

The funders had no role in study design, data collection and interpretation, or the decision to submit the work for publication.

### Author contributions

Typhaine Comyn, Conceptualization, Investigation, Methodology, Writing – original draft, Writing – review and editing; Thomas Preat, Conceptualization, Funding acquisition, Writing – review and editing; Alice Pavlowsky, Conceptualization, Supervision, Methodology, Writing – original draft, Writing – review and editing; Pierre-Yves Plaçais, Conceptualization, Supervision, Funding acquisition, Methodology, Writing – original draft, Writing – review and editing

### Author ORCIDs

Alice Pavlowsky (iD) https://orcid.org/0000-0001-6873-0577
Pierre-Yves Plaçais (iD) https://orcid.org/0000-0001-8426-4465

Reviewer #1 (Public review): https://doi.org/10.7554/eLife.92085.3.sa1
Reviewer #2 (Public review): https://doi.org/10.7554/eLife.92085.3.sa2
Author response https://doi.org/10.7554/eLife.92085.3.sa3

## Additional files

### Supplementary files
• MDAR checklist

### Data availability
All data generated or analysed during this study are included in the manuscript and supporting files; source data files have been provided that contain numerical data used to generate all figures.

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
