## [Editor Report · eLife Assessment]

This is a **fundamental** research study which identifies some of the molecular mechanisms underlying the energy costly process of memory consolidation. The strength of evidence is **exceptional**. The paper should be of broad interest because it establishes a clear mechanistic link between long-term memory processes and the energy-producing machinery in neurons.

---

## [Referee Report · Reviewer #1 (Public review)]

Summary:

This is a detailed description of the role of PKCδ in *Drosophila* learning and memory. The work is based on a previous study (Placais et al. 2017) that has already shown that for the establishment of long-term memory, the repetitive activity of MP1 dopaminergic neurons via the dopamine receptor DAMB is essential to increase mitochondrial energy flux in the mushroom body. In this paper, the role of PKCδ is now introduced. PKCδ is a molecular link between the dopaminergic system and the mitochondrial pyruvate metabolism of mushroom body Kenyon cells. For this purpose, the authors establish a genetically encoded FRET-based fluorescent reporter of PKCδ-specific activity, δCKAR.

Strengths:

This is a thorough study on the long-term memory of *Drosophila*. The work is based on the extensive, high-quality experience of the senior authors. This is particularly evident in the convincing use of behavioral assays and imaging techniques to differentiate and explore various memory phases in *Drosophila*. The study also establishes a new reporter to measure the activity of PKCδ - the focus of this study - in behaving animals. The authors also elucidate how recurrent spaced training sessions initiate a molecular gating mechanism, linking a dopaminergic punishment signal with the regulation of mitochondrial pyruvate metabolism. This advancement will enable a more precise molecular distinction of various memory phases and a deeper comprehension of their formation in the future.

Weaknesses:

The study offers novel insights into the molecular mechanisms underlying long-term memory formation and presents no apparent weaknesses in either content or methodology.

---

## [Referee Report · Reviewer #2 (Public review)]

Summary

This study deepens the former authors' investigations of the mechanisms involved in gating the long-term consolidation of an associative memory (LTM) in *Drosophila melanogaster*. After having previously found that LTM consolidation 1. costs energy (Plaçais and Préat, Science 2013) provided through pyruvate metabolism (Plaçais et al., Nature Comm 2017) and 2. is gated by the increased tonic activity in a type of dopaminergic neurons ('MP1 neurons') following only training protocol relevant for LTM, i.e. interspaced in time (Plaçais et al., Nature Neuro 2012), they here dig into the intra-cell signalling triggered by dopamine input and eventually responsible for the increased mitochondria activity in Kenyon Cells. They identify a particular PKC, PKCδ, as a major molecular interface in this process and describe its translocation to mitochondria to promote pyruvate metabolism, specifically after spaced training.

Methodological approach

To that end, they use RNA interference against the isozyme PKCδ, in a time-controlled way and in the whole Kenyon cells populations or in the subpopulation forming the α/β lobe. This knock-down decreased the total PKCδ mRNA level in the brain by ca. 30%, and is enough to observe decreased in flies performances for LTM consolidation. Using Pyronic, a sensor for pyruvate for in vivo imaging, and pharmacological disruption of mitochondrial function, the authors then show that PKCδ knock-down prevents high level of pyruvate from accumulating in the Kenyon cells at the time of LTM consolidation, pointing towards a role of PKCδ in promoting pyruvate metabolism. They further identify the PDH kinase PDK as a likely target for PKCδ since knocking down both PKCδ and PDK led to normal LTM performances, likely counterbalancing PKCδ knock-down alone.

To understand the timeline of PKCδ activation and to visualise its mitochondrial translocation in subpart of Mushroom body lobes they imported in fruitfly the genetically-encoded FRET reporters of PKCδ, δCKAR and mitochondria-δCKAR (Kajimoto et al 2010). They show that PKCδ is activated to the sensor's saturation only after spaced training, and not other types of training that are 'irrelevant' for LTM. Further, adding thermogenetic activation of dopaminergic neurons and RNA interference against Gq-coupled dopamine receptor to FRET imaging, they identify that a dopamine-triggered cascade is sufficient for the elevated PKCδ-activation.

Strengths and weaknesses

The authors use a combination of new fluorescent sensors and behavioral, imaging, and pharmacological protocols they already established to successfully identify the molecular players that bridge the requirement for spaced training/dopaminergic neurons MP1 oscillatory activity and the increased metabolic activity observed during long-term memory consolidation.

The study is dense in new exciting findings and each methodological step is carefully designed. The experiments one could think of to make this link have been done in this study and the results seem solid.

The discussion is well conducted, with interesting parallel with mammals, where the possibility that this process takes place as well is yet unknown.

Impact

Their findings should interest a large audience:

They discover and investigate a new function for PKCδ in regulating memory processes in neurons in conjunction with other physiological functions, making this molecule a potentially valid target for neuropathological conditions. They also provide new tools in *Drosophila* to measure PKCδ activation in cells. They identify the major players for lifting the energetic limitations preventing the formation of a long-term memory.

---

## [Author Response]

The following is the authors’ response to the original reviews.

**Public Reviews:**

**Reviewer #1 (Public Review):**
Summary:This is a detailed description of the role of PKCδ in *Drosophila* learning and memory. The work is based on a previous study (Placais et al. 2017) that has already shown that for the establishment of long-term memory, the repetitive activity of MP1 dopaminergic neurons via the dopamine receptor DAMB is essential to increase mitochondrial energy flux in the mushroom body.In this paper, the role of PKCδ is now introduced. PKCδ is a molecular link between the dopaminergic system and the mitochondrial pyruvate metabolism of mushroom body Kenyon cells. For this purpose, the authors establish a genetically encoded FRET-based fluorescent reporter of PKCδspecific activity, δCKAR.Strengths:This is a thorough study of the long-term memory of *Drosophila*. The work is based on the extensive, high-quality experience of the senior authors. This is particularly evident in the convincing use of behavioral assays and imaging techniques to differentiate and explore various memory phases in *Drosophila*. The study also establishes a new reporter to measure the activity of PKCδ - the focus of this study - in behaving animals. The authors also elucidate how recurrent spaced training sessions initiate a molecular gating mechanism, linking a dopaminergic punishment signal with the regulation of mitochondrial pyruvate metabolism. This advancement will enable a more precise molecular distinction of various memory phases and a deeper comprehension of their formation in the future.Weaknesses:Apart from a few minor technical issues, such as the not entirely convincing visualisation of the localisation of a PKCδ reporter in the mitochondria, there are no major weaknesses. Likewise, the scientific classification of the results seems appropriate, although a somewhat more extensive discussion in relation to *Drosophila* would have been desirable.

We are very grateful for this very positive appreciation of our work. Following this comment, we have revised our manuscript to bring more compelling evidence of the mitochondrial localization of the PKCδ reporter. We also developed the discussion of our results with respect to the *Drosophila* learning and memory literature.

**Reviewer #2 (Public Review):**
SummaryThis study deepens the former authors' investigations of the mechanisms involved in gating the longterm consolidation of an associative memory (LTM) in *Drosophila melanogaster*. After having previously found that LTM consolidation 1. costs energy (Plaçais and Préat, Science 2013) provided through pyruvate metabolism (Plaçais et al., Nature Comm 2017) and 2. is gated by the increased tonic activity in a type of dopaminergic neurons ('MP1 neurons') following only training protocol relevant for LTM, i.e. interspaced in time (Plaçais et al., Nature Neuro 2012), they here dig into the intra-cell signalling triggered by dopamine input and eventually responsible for the increased mitochondria activity in Kenyon Cells. They identify a particular PKC, PKCδ, as a major molecular interface in this process and describe its translocation to mitochondria to promote pyruvate metabolism, specifically after spaced training.Methodological approachTo that end, they use RNA interference against the isozyme PKCδ, in a time-controlled way and in the whole Kenyon cell populations or in the subpopulation forming the α/β lobe. This knock-down decreased the total PKCδ mRNA level in the brain by ca. 30%, and is enough to observe decreased in flies performances for LTM consolidation. Using Pyronic, a sensor for pyruvate for in vivo imaging, and pharmacological disruption of mitochondrial function, the authors then show that PKCδ knockdown prevents a high level of pyruvate from accumulating in the Kenyon cells at the time of LTM consolidation, pointing towards a role of PKCδ in promoting pyruvate metabolism. They further identify the PDH kinase PDK as a likely target for PKCδ since knocking down both PKCδ and PDK led to normal LTM performances, likely counterbalancing PKCδ knock-down alone.To understand the timeline of PKCδ activation and to visualise its mitochondrial translocation in a subpart of Mushroom body lobes they imported in fruitfly the genetically-encoded FRET reporters of PKCδ, δCKAR, and mitochondria-δCKAR (Kajimoto et al 2010). They show that PKCδ is activated to the sensor's saturation only after spaced training, and not other types of training that are 'irrelevant' for LTM. Further, adding thermogenetic activation of dopaminergic neurons and RNA interference against Gq-coupled dopamine receptor to FRET imaging, they identify that a dopamine-triggered cascade is sufficient for the elevated PKCδ-activation.Strengths and weaknessesThe authors use a combination of new fluorescent sensors and behavioral, imaging, and pharmacological protocols they already established to successfully identify the molecular players that bridge the requirement for spaced training/dopaminergic neurons MP1 oscillatory activity and the increased metabolic activity observed during long-term memory consolidation.The study is dense in new exciting findings and each methodological step is carefully designed. Almost all possible experiments one could think of to make this link have been done in this study, with a few exceptions that do not prevent the essential conclusions from being drawn.The discussion is well conducted, with interesting parallels with mammals, where the possibility that this process takes place as well is yet unknown.ImpactTheir findings should interest a large audience:They discover and investigate a new function for PKCδ in regulating memory processes in neurons in conjunction with other physiological functions, making this molecule a potentially valid target for neuropathological conditions. They also provide new tools in *Drosophila* to measure PKCδ activation in cells. They identify the major players for lifting the energetic limitations preventing the formation of a long-term memory.

We warmly thank Reviewer #2 for the enthusiastic assessment of our work. There were no specific point to address in the Public Review.

**Recommendations for the authors:**

**Reviewer #1 (Recommendations For The Authors):**
I have a few comments that could help improve the paper and help the reader navigate the detailed analysis.(1) Perhaps the authors could add a sentence or two in the intro about the different PKC genes in *Drosophila* and whether they are expressed in the MB.

We thank Reviewer #1 for this suggestion. We now describe in the introduction the various subfamilies of PKCs downstream of Gq signaling , the *Drosophila* members of those different PKC subfamilies, and their expression in the brain.

(2) Italicise *Drosophila* throughout the text.

We have done this correction.

(3) In Figure 1, you could change the scheme in Figure F-H and have the timeline always start after training. Then you could see that the training varies in time (perhaps provide the exact duration for each training protocol) and the test interval is constant. Why is it actually measured in a time window and not at an exact time?

This is indeed a good suggestion to clarify the presentation of our results. We changed the timelines schemes in all the figures with the t=0 starting at the end of the conditioning. Indeed, each conditioning protocol has a different duration as represented on these timelines: as one-cycle training lasts 5 min, 5x massed training has a duration of 20 min, and 5x spaced training takes 1 hours and 30 min to be completed, with its 15 min intertrial intervals. In vivo imaging experiments are performed during a certain time window after conditioning during which, according to our previous experience, the activity of MP1 dopamine neurons after spaced training remains constant (Plaçais et al., 2012). This offers the practical advantage that we can image several flies after a given training session, instead of having to perform many consecutive conditioning protocols.

(4) In Figure 2 you could show the massed training data from the supplement. This is very similar to what is shown in Figure 1. Are there also imaging experiments on massed training?

The reason why massed training data was initially displayed in the supplementary data is that α/β neurons are known to be crucial for LTM formation but are not required for memory formed after massed training, so that the absence of effect was somehow expected. Nonetheless, we performed δCKAR imaging in α/β neurons after 5x massed training and found that PKCδ activity was not increased post-conditioning as expected (Figure 2C). This experiment was performed in parallel of additional data after 5x spaced conditioning δCKAR imaging in α/β neurons as a positive control (these new data were added to the Figure 2B). Following Reviewer #1’s suggestion, all data investigating the effect of PKCδ in α/β neurons are now displayed on Figure 2.

(5) Figure 3: I am not sure if the blue curve in Figure A really represents an upregulated pyruvate flux compared to the control (mentioned in line 210). It may be the case initially, but it is clearly below the control after 40s. Why is that?

This visual effect is due to the fact that PDBu injection in itself increases the pyruvate level in MB neurons (independently of its effect on PKCδ), before sodium azide injection. As a result, the baseline of the PDBu treated flies is above the DMSO control flies when sodium azide is injected, which results in the fact that the pyronic sensor saturates quicker and therefore reaches its plateau before the control when traces where normalized right before sodium azide injection.

That being said, the measure of the slope in itself following sodium azide injection is not affected by these differences, and is always measured between 10 and 70% of the plateau.

Given this remark, and another comment from Reviewer#2 about this experiment, we removed the panel 3A and present only the complete recording of this experiment, that is now displayed on Figure 3 – figure supplement 1C.

(6) For me, the localisation of the mitochondrial reporter in the mitochondria is not clear. The image in the supplement is not sufficient to show this clearly. What is missing here is a co-staining in the same brain of UAS-mito-δCKAR and a mitochondrial marker to label the mitochondria and the reporter at the same time in the same animal.

We agree with Reviewer #1’s remark and added new data to make this point more convincing. As suggested, we co-expressed mito-δCKAR with the mitochondrial reporter mito-DsRed in MB neurons (Lutas et al., 2012). We observed a clear colocalization of both signals by performing confocal imaging in the MB neurons somas, indicating that mito-δCKAR is indeed addressed to mitochondria (Figure 4 – figure supplement 1B and 2).

(7) Are there controls that the MB expression of the reporters in the flies does not influence the learning ability? In order to make statements about the physiology of the cells, it must also be shown that the cells still have normal activity and allow learning behaviour comparable to wild-type flies.

This is indeed an important control that we added in the revised version. We tested the memory after 5x spaced, 5x massed and 1x training of flies expressing in the MB the various imaging probes used in our study (cyto-δCKAR, mito-δCKAR and Pyronic). Memory performance was similar to controls in all cases (Figure 1 – figure supplement 1E).

(8) Perhaps the authors could go into more detail on two points in the discussion and shorten the comprehensive comparison to the vertebrate system somewhat. It would be nice to know how the local transfer from the peduncle to the vertical lobus is supposed to take place. What is the mechanism here? Any suggestions from the literature? It would also be useful to mention the compartmentalisation of the MB and how the information can overcome these boundaries from the peduncle to the vertical lobe.

We now elaborate on this question in the discussion (lines 368-386). To sum up, given that the compartmentalization of the MBs is anatomically defined by the presence of specific subset of MBON and DAN cell types (forming different information-processing units), rather than by physical boundaries per se, we can consider two main hypotheses to explain PKCδ activation transfer from the peduncle to the lobes: passive diffusion of activated PKCδ, or mitochondrial motility that would displace PKCδ from its place of first activation. We indeed found that mitochondrial motility was occurring upon 5x spaced conditioning for LTM formation (Pavlowsky et al. 2024).

In principle, one could also consider that PKCδ could be activated in the lobes by a relaying neuron. The MVP2 neuron (aka MBON-γ1>pedc) presents dendrites facing MP1 and makes synapses with the α/β neurons at the level of the α and β lobes, which makes it a good candidate. Furthermore, as we show that PKCδ activation in the lobes requires DAMB (Figure 4C, Figure 5A-B, Figure 5 – figure supplement 1), one could imagine the following activation loop: MP1 activates the MB neurons via DAMB, that activate MVP2 at the level of the peduncle, which activates in turn the MB neurons at the level of the lobes. However, we did not retain this hypothesis, because MVP2 is GABAergic, which makes it highly unlikely to be able to activate a kinase like PKCδ.

Regarding the comparative discussion with mammalian systems, we appreciate Reviewer #1’s remark that it may appear too detailed, but given that Reviewer #2 (public comment) highlighted the ‘interesting parallel with mammals’ in our discussion, we finally chose not to reduce this part in the revised manuscript.

**Reviewer #2 (Recommendations For The Authors):**
Fig 1G: is there a decrease in PKCδ activation after mass training as compared to the control, indicating an inhibitory mechanism onto PKCδ following mass training? Or is this an artifact of the PDBu application procedure in the control group?

We thank Reviewer #2 for this careful comment. The dent in the timetrace following PDBu application after massed training (Figure 1G) is indeed an artifact due to the manual injection of the drug. But we would like to emphasize that what matters in the determination of PKCδ activity is the level of the baseline before PDBu application after normalization to the final plateau, so that variation around the injection time do not impact the result of the analysis. Moreover, in the revised version, we performed a similar series of experiments, using an α/β neuron-specific driver (Figure 2C). In this series of experiments, there were limited injection artefacts, and we obtained the same conclusion as Figure 1G that PKCδ activity is left unchanged by 5x massed conditioning.

Fig 3A: I suggest moving this panel in the supplement: I found it difficult to process the effect of PDBu that is unspecific to PKCδ and that leads to a different plateau because of a different baseline. It would be better explained in more detail in the supplement, especially given that the 3B panel can lead to a similar conclusion and does not have this specificity problem. Up to the authors.

We thank Reviewer #2 for this feedback. We followed the suggestion and now only display the full recording of this experiment on Figure 3 – figure supplement 1C.

Fig 3C: To go further, one wonders if knocking-down PDK would act as a switch for gating LTM formation, i.e. if done during a 1x training or a 5x massed training would it gate long-term consolidation?

This is indeed an excellent suggestion. We performed this experiment and showed that in flies expressing the PDK RNAi in adult MB neurons, only one cycle of training was sufficient to induce longterm memory formation (Figure 3A), instead of the 5 spaced cycles normally required. This confirms the model we previously established in Plaçais et al. 2017, where long-term memory formation was observed upon PDK MB knock-down after 2 cycles of spaced training. This new result goes further in characterizing this facilitation effect, now showing that even a single cycle is sufficient. Altogether these data show that mitochondrial metabolic activation is the critical gating step in long-term memory formation. Spaced training achieves this activation through PDK inhibition, mediated by PKCδ.

What is the level of mRNA in this construct? I don't see a quantification, can you justify it?

We thank Reviewer #2 for this remark. This PDK RNAi had been used in a previous work in pyruvate imaging experiment, where it successfully boosted mitochondrial pyruvate uptake. But indeed we had not validated it at the mRNA level. In the revised version of the present manuscript, we now confirm by RT-qPCR that the PDK RNAi efficiently downregulates PDK expression in neurons (Figure 3 – figure supplement 1A).

Fig. 4C: Is PKCδ activation increase in Vertical lobe DAMB-dependent? One wonders, because MP1 may somehow activate other neurons that could reach this part of the Kenyon Cells. I do not see in the results what could disprove this possibility. The mechanism linking DAMB activation in the peduncle and PKCδ activation in the VL is mysterious, see also Fig. 5.

This is a very sound remark. In the revised version we have checked whether PKCδ activation in the vertical lobes is also dependent on DAMB. We performed thermogenetic activation of MP1 neurons and imaged mito-δCKAR signal in the vertical lobes upon DAMB MB knock-down. We found that as for the peduncle, DAMB was required for PKCδ mitochondrial activation (Figure 4C, right panel). This experiment was performed in parallel with similar measurements in flies that did not express DAMB RNAi, as a positive control (these new control data were added to the Figure 4C, left panel).

This result supports a model where dopamine from MP1 neurons directly acts on Kenyon cells, even for PKCδ activation in the vertical lobes. Thus, this advocates for a diffusion of DAMB-activated PKCδ from the peduncle to the vertical lobes, either by passive diffusion or by mitochondrial motility - two hypotheses that we added in the discussion.

Fig. 5: If MP1 neurons release dopamine only to the peduncle, how do you expect PKCδ to be translocated to mitochondria all the way to the vertical lobe? Also is it specific to the vertical lobe and not found in the medial lobe?

Investigating the spatial distribution of PKCδ is, once again, a very sound suggestion. We re-analyzed our dataset of the mito-δCKAR signal after spaced training for peduncle measurement, as the imaging plane also included the β lobe. We found that PKCδ is also activated at that level, and that its activation also depends on DAMB (Figure 5 – figure supplement 1). We also performed additional pyruvate measurements in the medial lobes, and observed that mitochondria pyruvate uptake presents the same extension in time in the medial lobes as in the vertical lobes when comparing spaced training (Figure 6 E-F and Figure 6 – figure supplement 1E-F) to 1x training (Figure 6A-B and Figure 6 – figure supplement 1C-D). Therefore, the metabolic action of PKCδ seems not to be restricted to the vertical lobes, but spreads across the whole axonal compartment.

Altogether, these data point toward the fact that activated PKCδ diffuse from its point of activation, the peduncle, where dopamine is released by MP1 and DAMB is activated, to both the vertical and medial lobes, either by passive diffusion, or taking advantage of mitochondrial movement that was shown to be triggered by spaced training (Pavlowsky et al. 2024), from the MB neurons somas to the axons. To further characterize the kinetics of PKCδ activation, we measured its activity using the mitoδCKAR sensor at 3 and 8 hours following spaced training. We found that while PKCδ was still active at 3 hours, it was back to its baseline activity level at 8 hours, both at the level of the peduncle and the vertical lobes (Figure 5 C-F). However, at 8 hours, pyruvate metabolism is still upregulated in the lobes, which indicates that an additional mechanism is relaying PKCδ action to maintain the high energy state of the MBs at later time points. As we propose in the revised discussion, the mitochondrial motility hypothesis makes sense here (Pavlowsky et al. 2024), as the progressive increase in the number of mitochondria in the lobes would be able to sustain high mitochondrial metabolism beyond PKCδ activation at 8 hours post-conditioning. This new result and its implications open exciting perspectives for future research about the different mitochondrial regulations occurring after spaced training, their organization over time and their interactions.

Fig.7: PDK written in yellow is almost invisible

This has been changed.